# Recruitment of Ahsa1 to Hsp90 is regulated by a conserved peptide that inhibits ATPase stimulation

Solomon K Hussein [1], Rakesh Bhat [2], Michael Overduin[2] & Paul LaPointe [1]✉

## Abstract

Hsp90 is a molecular chaperone that acts on its clients through an ATP-dependent and conformationally dynamic functional cycle. The cochaperone Accelerator of Hsp90 ATPase, or Ahsa1, is the most potent stimulator of Hsp90 ATPase activity. Ahsa1 stimulates the rate of Hsp90 ATPase activity through a conserved motif, NxNNWHW. Metazoan Ahsa1, but not yeast, possesses an additional 20 amino acid peptide preceding the NxNNWHW motif that we have called the intrinsic chaperone domain (ICD). The ICD of Ahsa1 diminishes Hsp90 ATPase stimulation by interfering with the function of the NxNNWHW motif. Furthermore, the NxNNWHW modulates Hsp90's apparent affinity to Ahsa1 and ATP. Lastly, the ICD controls the regulated recruitment of Hsp90 in cells and its deletion results in the loss of interaction with Hsp90 and the glucocorticoid receptor. This work provides clues to how Ahsa1 conserved regions modulate Hsp90 kinetics and how they may be coupled to client folding status.

**Keywords** Hsp90; Ahsa1/Aha1; Chaperones; Protein Folding; ATPase
**Subject Category** Translation & Protein Quality

## Introduction

The 90-kDa heat shock protein (Hsp90) is a molecular chaperone responsible for aiding the folding of hundreds of substrates called clients (Echeverria et al, 2011; Eckl and Richter, 2013; Taipale et al, 2012; Van Oosten-Hawle et al, 2017). Hsp90 clients include transcription factors, kinases, membrane proteins, and many other types of proteins (Echeverria et al, 2011; Taipale et al, 2012; Taipale et al, 2014). Hsp90 processes client proteins by progression through a conformationally dynamic functional cycle. The in vivo action of Hsp90 is regulated by a set of proteins called cochaperones as well as numerous post-translational modifications (PTMs) that can modulate cochaperone and client interactions, as well as the kinetics of the functional cycle (Armstrong et al, 2012; Chang et al, 1997; Fang et al, 1998; Knoblauch and Garabedian, 1999; Lee et al, 2012; Li et al, 2013; Li et al, 2012; Mollapour et al, 2014; Mollapour and Neckers, 2012; Mollapour et al, 2010; Mollapour et al, 2011a;

Mollapour et al, 2011b; Nathan et al, 1999; Panaretou et al, 2002; Prodromou et al, 1999; Richter et al, 2004; Sager et al, 2019; Siligardi et al, 2004; Soroka et al, 2012; Woodford et al, 2016; Woodford et al, 2017; Zuehlke et al, 2017).

At the heart of the Hsp90 functional cycle is the binding and hydrolysis of ATP by this dimeric chaperone (Pearl, 2016; Prodromou et al, 2000; Richter et al, 2008). While it has recently been suggested that ATP hydrolysis by Hsp90 is not absolutely essential (Reidy et al, 2023; Zierer et al, 2016), it is clear that timely binding and hydrolysis is important for optimal Hsp90 activity (Obermann et al, 1998; Panaretou et al, 1998). The most potent stimulator of the normally very low ATPase activity of Hsp90 is the Accelerator of Hsp90 ATPase (Aha1)(Horvat et al, 2014; Lotz et al, 2003; Meyer et al, 2004; Panaretou et al, 2002). Ahsa1, the human homolog of Aha1, has been linked to several Hsp90 client proteins that play central roles in human disease (Shelton et al, 2017; Wang et al, 2006). Silencing Ahsa1 expression promotes the folding of the cystic fibrosis transmembrane conductance regulator (CFTR) (Wang et al, 2006). Moreover, silencing Ahsa1 expression also restores the folding of the most common disease-associated variant of CFTR, ΔF508. Ahsa1 has also been linked to Alzheimer's disease because of its ability to influence the aggregation of Tau (Shelton et al, 2017). Tau tangles and neurofibrillary aggregates are hallmarks of Alzheimer's disease etiology. Overexpression of Ahsa1 in mouse brains is reported to enhance Tau aggregation and neuronal cell death. Ahsa1 expression has also been shown to influence the growth of human tumor cells and their sensitivity to ATP-competitive Hsp90 inhibitors (Holmes et al, 2008). Of note, a homolog of Aha1 in yeast, Hch1, regulates sensitivity to Hsp90 inhibitors (Armstrong et al, 2012).

A recent study of chaperones and cochaperones across the tree of life has shown that Hsp90 cochaperones (i.e. Aha1 and many others) appeared first in unicellular eukaryotes (Rebeaud et al, 2021). Indeed, searches for sequences with high similarity to Ahsa1 reveal Ahsa1 orthologues in plants, fungi, insects, fish, mammals, etc. Human Ahsa1, like its counterpart in yeast, has two domains joined by a flexible linker (Fig. 1A). The NxNNWHW and RKxK motifs in the N-domain are the most conserved regions in Aha-type cochaperones. Both motifs are important for Aha1 function and maximal Hsp90 ATPase stimulation. In yeast, the NxNNWHW is found in the first 11 amino acids of the protein, and its deletion impairs ATPase stimulation of Hsp90 and the function of Aha1 in vivo (Mercier et al, 2019). However, Ahsa1 in all species we have examined possess an additional 20 amino acid sequence preceding the NxNNWHW motif. (Fig. 1B). In addition, only Aha1 from

[1]Department of Cell Biology, Faculty of Medicine & Dentistry, University of Alberta, Edmonton, Alberta T6G 2H7, Canada. [2]Department of Biochemistry, Faculty of Medicine & Dentistry, University of Alberta, Edmonton, Alberta T6G 2H7, Canada. ✉E-mail: paul.lapointe@ualberta.ca

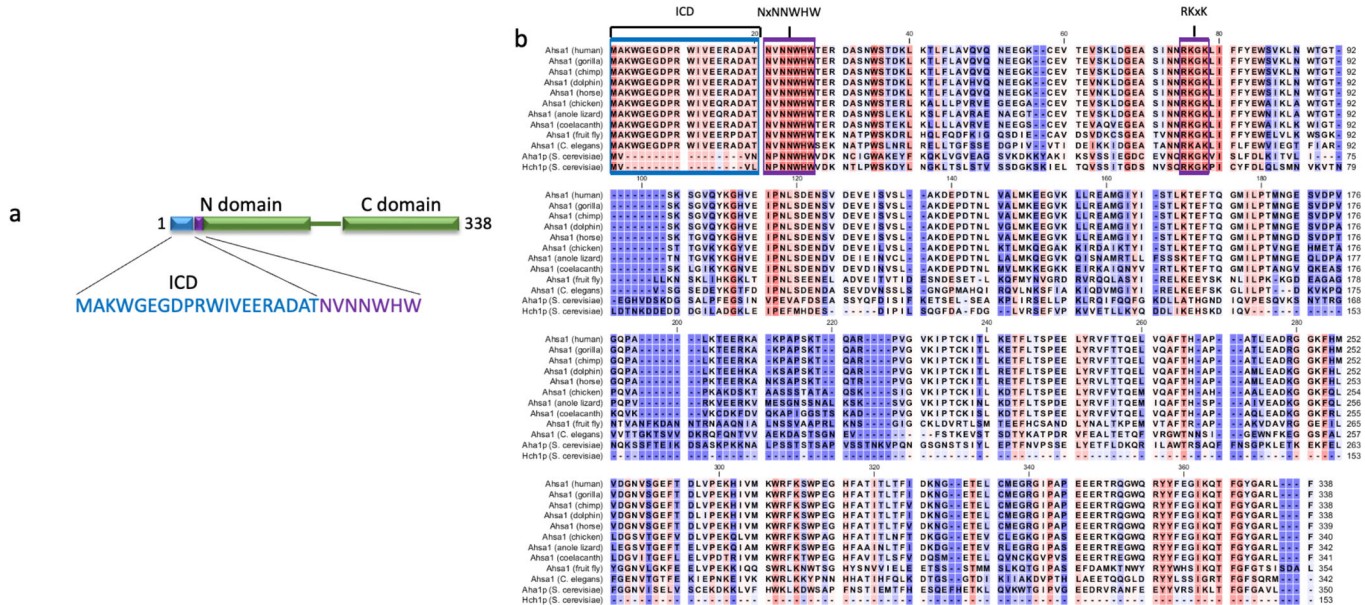

**Figure 1. The first 20 amino acids of Ahsa1 cochaperones are conserved in all organisms but *Saccharomyces cerevisiae*.**

(A) A schematic diagram of human Ahsa1 showing the NxNNWHW motif occurring after a conserved 20 amino acid sequence previously linked to intrinsic chaperone activity (referred to as the Intrinsic Chaperone Domain (ICD)). (B) The ICD is highly conserved in all non-yeast Ahsa1 proteins.

fungi lacks the 20 amino acid sequence present in human Ahsa1. In fact, this sequence is just as strongly conserved as the NxNNWHW and RKxK motifs that characterize the Aha cochaperone family (Fig. 1B). Despite the presence of this strongly conserved sequence across organisms that possess an Aha orthologue, very little has been done to determine its significance. Although, intrinsic chaperone function has been attributed to this 20 amino acid sequence in both in vitro protein aggregation and in vivo folding assays (Liu and Wang, 2022; Tang et al, 2023; Tripathi et al, 2014).

We report here that the 20 amino acid intrinsic chaperone domain (ICD) of Ahsa1 diminishes Hsp90 ATPase stimulation activity. Specifically, the ICD partially interferes with the participation of the NxNNWHW motif that is immediately downstream. Moreover, in cells, the loss of the ICD prevents the interaction of Ahsa1 with Hsp90 and the Hsp90 client protein, the glucocorticoid receptor (GR). While human Ahsa1 is distinct from yeast Aha1 with respect to the ICD and in vivo recruitment to Hsp90, we show that the NxNNWHW motif plays a similar role in both. Specifically, it regulates the apparent affinity for ATP and the rate of ATP hydrolysis.

This work provides a framework to understand the conserved functions of ATPase stimulating cochaperones and provides clues regarding how Ahsa1 function has diverged in metazoans to be coupled to client folding status.

## Results

### Deletion of the ICD in Ahsa1 enhances ATPase stimulation of Hsp90

Previous reports demonstrated that the metazoan-specific N terminal extension (which we term here the ICD) on Ahsa1

imparts intrinsic chaperone activity to this cochaperone (Liu and Wang, 2022; Tang et al, 2023; Tripathi et al, 2014). In the 2014 study, deletion of the first 22 amino acids resulted in a loss of the intrinsic chaperone activity but did not impair ATPase stimulation of Hsp90 (Tripathi et al, 2014). However, we noticed that the 22 amino acid truncation used in that study included the ICD (20 amino acids) and residues of the NxNNWHW motif that we recently discovered to be required for maximal stimulation of yeast Hsp90 by Aha1 (Mercier et al, 2019). To separate the effect of deleting the 20 amino acid ICD from that of the NxNNWHW motif, we expressed and purified a series of Ahsa1 constructs for testing in an in vitro ATPase stimulation assay (Fig. 2A). We expressed full-length Ahsa1, Ahsa1$^{\Delta 20}$ (missing the entire sequence upstream of the NxNNWHW), and Ahsa1$^{\Delta 27}$ (lacking both the ICD and the NxNNWHW motif), all harboring a C-terminal poly-histidine (6xHis) tag. Interestingly, the maximal ATPase stimulation ($B_{max}$) of both Hsp90α and Hsp90β was dramatically higher with Ahsa1$^{\Delta 20}$ than with full-length Ahsa1 (Fig. 2B–D). Further deletion of the NxNNWHW motif reduced ATPase stimulation of Hsp90α and Hsp90β to the same level as full-length Ahsa1. This result was consistent with our previous work with yeast Aha1 where we showed that removal of the NxNNWHW motif impaired ATPase stimulation (Mercier et al, 2019). In addition, while the $K_{app}$ of Ahsa1 for Hsp90 was unchanged after deletion of the ICD, further deletion of the NxNNWHW motif reduced the apparent affinity of Ahsa1 for both Hsp90α and Hsp90β (Fig. 2E).

### Deletion of the NxNNWHW motif of Ahsa1 enhances the apparent affinity for ATP of Hsp90α and Hsp90β

We previously showed that deletion of the NxNNWHW motif in yeast Aha1 altered the apparent affinity for ATP of Hsp90 (Mercier

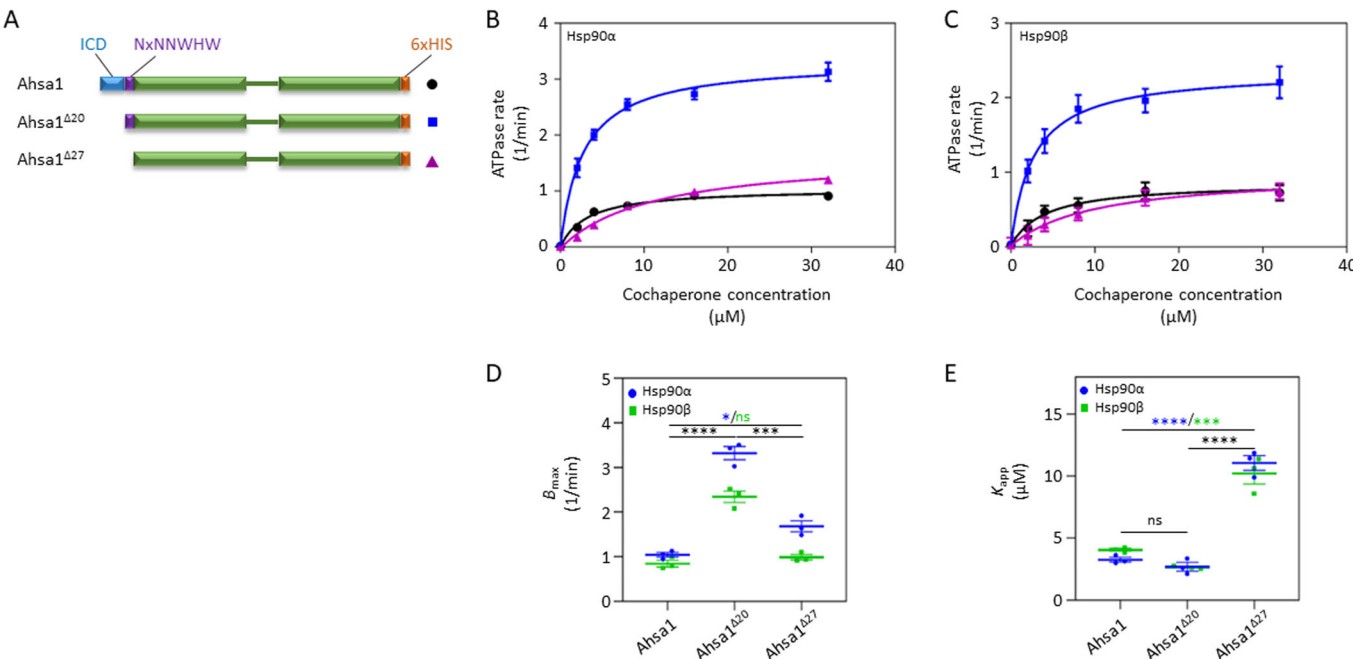

**Figure 2. The Ahsa1 ICD modulates ATPase stimulation of Hsp90.**

ATPase stimulation of Hsp90α and Hsp90β by Ahsa1 constructs (A). Schematic of Ahsa1 constructs used. (B, C) Stimulation of Hsp90α (B) and Hsp90β (C) ATPase activity by increasing concentrations of full-length Ahsa1 (black circles), Ahsa1$^{\Delta20}$ (blue squares), and Ahsa1$^{\Delta27}$ (purple triangles). (D) $B_{max}$ values of each of the three replicate reactions shown in (A) blue) and (B) (green). (E) $K_{app}$ values of the reactions shown in (A) (blue) and (B) (green). Reactions contained 2 μM Hsp90α or Hsp90β and indicated concentration of cochaperone. $N = 3$. Data Information: In (B, C), data for each concentration of cochaperone are presented as mean $+/-$ SEM of three independent experiments ($N = 3$; each $N$ is one experiment carried out with technical triplicates as described in "Methods"). (D, E) The individual $B_{max}$ or $K_{app}$ values, respectively, are plotted as well as the mean $+/-$ SEM of the three independent experiments shown in (B, C) ($N = 3$; each $N$ is one experiment carried out with technical triplicates as described in "Methods"). Statistical significance in (D, E) was calculated using a Tukey's multiple comparisons test (*$P \leq 0.05$; ***$P \leq 0.001$; ****$P \leq 0.0001$) ($P$ value symbols in black apply to Hsp90α and Hsp90β; $P$ value symbols in blue apply to Hsp90α; $P$ value symbols in green apply to Hsp90β). Source data are available online for this figure.

et al, 2019). To see if this motif played a similar role in regulating affinity for nucleotide in the mammalian system, we tested our constructs in ATPase reactions with varying concentrations of ATP. We found that deletion of the ICD resulted in a small but statistically significant reduction in apparent $K_m$ for ATP. Subsequent deletion of the NxNNWHW motif further reduced the apparent $K_m$ for ATP, suggesting that this motif plays a similar role in mammalian Ahsa1 and yeast Aha1 (Fig. 3).

## The ICD of Ahsa1 partially masks the NxNNWHW motif

Our results show that the ICD inhibits the ability of Ahsa1 to stimulate the ATPase rate of Hsp90 (Fig. 2B,C). Given the similarity between the ATPase stimulation rates of Ahsa1 and Ahsa1$^{\Delta27}$, and the proximity of the ICD to the NxNNWHW, we hypothesized that the ICD might simply be masking the NxNNWHW motif in some way. This hypothesis predicts that mutations in the NxNNWHW motif would severely impact ATPase stimulation in the context of the Ahsa1$^{\Delta20}$ construct but have no effect in the context of full-length Ahsa1. To test this, we introduced point mutations into the NxNNWHW in both full-length Ahsa1 and Ahsa1$^{\Delta20}$ and tested them in ATPase assays with both Hsp90α and Hsp90β (Fig. EV1). Mutation of all three asparagine residues in the NxNNWHW motif of full-length Ahsa1 caused a statistically significant reduction in the $B_{max}$ for both

Hsp90α (Fig. 4A) and Hsp90β (Fig. 4B). In contrast, all mutations in the NxNNWHW motif in the Ahsa1$^{\Delta20}$ construct caused a statistically significant reduction in the $B_{max}$ for both Hsp90α (Fig. 4C) and Hsp90β (Fig. 4D). When we calculated the percent reduction in ATPase stimulation ($B_{max}$) of each mutant relative to the parental construct, we observed that all of the mutants we tested had a greater impact in the context of Ahsa1$^{\Delta20}$, but this was most significant for mutations in the WHW portion of the motif (Fig. 4E,F). This suggested that the ICD partially masks the NxNNWHW but must also inhibit ATPase stimulation by Ahsa1 by some other mechanism as well.

## Mutations in the NxNNWHW motif alter the apparent affinity for Hsp90 as well as for ATP

As shown above, the apparent affinity of the Ahsa1$^{\Delta27}$ construct for Hsp90 was significantly lower than that of full-length Ahsa1 or Ahsa1$^{\Delta20}$ (Fig. 2D). We observed a similar reduction in apparent affinity for both Hsp90α and Hsp90β in full-length Ahsa1 and Ahsa1$^{\Delta20}$ constructs harboring mutations in the WHW, but not the NxNN portion of the NxNNWHW motif (Fig. 5A–D). This suggested to us that the motif has separable roles in both binding to Hsp90 as well as in stimulation. Interestingly, mutations in the WHW portion of the NxNNWHW motif had an inverse effect on the apparent affinity for ATP (Fig. 6A,B). Only mutations in the

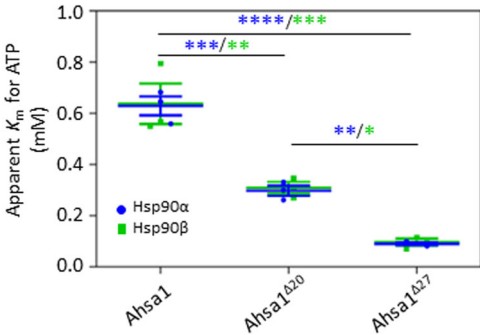

**Figure 3. The Ahsa1 ICD and NxNNWHW motif modulate the apparent affinity for ATP of Hsp90.**

Loss of the Ahsa1 ICD or loss of both ICD and NxNNWHW motif results in higher apparent affinity for ATP in both Hsp90α (blue) and Hsp90β (green). Kinetic analysis was carried out for Hsp90α or Hsp90β in the presence of full-length Ahsa1, Ahsa1$^{\Delta20}$, or Ahsa1$^{\Delta27}$. ATPase reactions were carried out with increasing concentrations of ATP (12.5, 25, 50, 100, 200, 400, 800, 1600 μM), and ATPase rates were analyzed by Michaelis–Menten nonlinear regression. Data Information: The apparent $K_m$ values for three independent experiments are plotted as well as the mean $+/-$ SEM ($N = 3$; each $N$ is one experiment carried out with technical triplicates as described in "Methods"). Statistical significance was calculated using a Tukey's multiple comparisons test ($*P \leq 0.05$; $**P \leq 0.01$; $***P \leq 0.001$; $****P \leq 0.0001$) ($P$ value symbols in blue apply to Hsp90α; $P$ value symbols in green apply to Hsp90β). Source data are available online for this figure.

WHW portion of the NxNNWHW motif led to significant differences on Hsp90 affinity for ATP.

## The ICD is necessary for stable interaction between Ahsa1 and Hsp90 in cells

Given the previously reported ability of Ahsa1 to interact with misfolded proteins in an Hsp90-independent manner, we wondered if the ICD played a role in the interaction with Hsp90. We transfected plasmids encoding either Ahsa1, Ahsa1$^{\Delta20}$, or Ahsa1$^{\Delta27}$ (each harboring a C-terminal myc tag) into MDA-MB-231 cells and immunoprecipitated each Ahsa1 construct with anti-myc beads. As expected, Hsp90 was recovered in complex with full-length Ahsa1. However, we did not recover any Hsp90 in complex with the Ahsa1$^{\Delta20}$ or Ahsa1$^{\Delta27}$ constructs (Fig. 7A). This was particularly surprising for the Ahsa1$^{\Delta20}$ construct, as there was no difference in affinity for Hsp90 in our in vitro ATPase assays (Fig. 2D). As we observed with Hsp90, the Hsp90 client glucocorticoid hormone receptor (GR) was recovered with full-length Ahsa1 but not with Ahsa1$^{\Delta20}$ or Ahsa1$^{\Delta27}$ (Fig. 7B). Quantification of Hsp90 and GR co-immunoprecipitation with the different Ahsa1 constructs was carried out using densitometry (Fig. EV2). This result was surprising because the apparent affinity of Ahsa1 and Ahsa1$^{\Delta20}$ for Hsp90 was identical in our in vitro ATPase assays. However, the apparent affinity only reports on the interaction of the first Ahsa1 binding event which is sufficient for ATPase stimulation so the interaction we observed in out IP experiments could be related to the differences in potential for 2:2 binding. To further investigate any binding defect associated with the deletion of the ICD, we measured the affinity of full-length Ahsa1 and Ahsa1$^{\Delta20}$ for Hsp90α directly using isothermal titration

calorimetry (ITC). We observed a relatively small difference in affinity (i.e. $K_D$ of 1.73 μM for Ahsa1 versus 3.20 μM for Ahsa1$^{\Delta20}$) (Fig. EV3). However, this difference seems unlikely to account for the total loss of binding we observed in our IPs.

## Interaction of Ahsa1 with Hsp90 is a precondition for complex formation with GR

Three studies have linked the ICD to direct client protein binding (Liu and Wang, 2022; Tang et al, 2023; Tripathi et al, 2014). This raises two possibilities with regard to the mechanism underlying complex formation between Ahsa1, Hsp90, and GR. One is that Ahsa1 may interact with clients through the ICD for subsequent delivery to Hsp90. The other is that Ahsa1 may interact with Hsp90-GR complexes that form first. To test this, we introduced the well-characterized E67K mutation into Ahsa1 that disrupts the interaction between the Ahsa1 N-domain and the Hsp90 middle domain (Heider et al, 2021; Liu et al, 2022; Meyer et al, 2004; Oroz et al, 2019; Shelton et al, 2017; Wang et al, 2006). We transfected plasmids encoding full-length Ahsa1, Ahsa1$^{E67K}$, or Ahsa1$^{\Delta20}$ (each harboring a C-terminal myc tag) into MDA-MB-231 cells and immunoprecipitated each Ahsa1 protein with anti-myc beads. As expected, Hsp90 was not recovered in complex with either Ahsa1$^{E67K}$ or Ahsa1$^{\Delta20}$ (Fig. 8A). However, GR was also no longer recovered with Ahsa1$^{E67K}$ suggesting that interaction with GR occurs after, or requires, binding to Hsp90 (Fig. 8B). Quantification of Hsp90 and GR co-immunoprecipitation with the different Ahsa1 constructs was carried out using densitometry (Fig. EV4).

## Discussion

The 20 amino acid sequence at the N-terminus of Ahsa1, despite its conservation in virtually every member of this cochaperone family, has been largely overlooked because it is absent in yeast Aha1. We report here that this sequence autoinhibits ATPase stimulation, in part by partially masking the NxNNWHW motif (Fig. 9A). More specifically, the ICD interferes with the WHW portion of the NxNNWHW motif but must also interfere with either another part of Ahsa1 or with Hsp90 directly. Deletion of these 20 amino acids results in a significant increase in Hsp90 ATPase stimulation. The only previous reports on this sequence suggested that it can interact with misfolded proteins directly and prevent aggregation (Liu and Wang, 2022; Tripathi et al, 2014). In one of these studies, a 22 amino acid deletion did not significantly increase ATPase stimulation of Hsp90. However, this could have been because part of the NxNNWHW was deleted in their construct. Indeed, we showed that the N21A substitution in Ahsa1 impaired Hsp90 ATPase stimulation. It is possible that the improvement in ATPase stimulation resulting from the deletion of the first twenty amino acids could have been masked by the deleterious effect of deleting the first two amino acids of the NxNNWHW motif. Additionally, we measured a far greater Ahsa1-stimulated Hsp90 ATPase rate (i.e. 3–5 fold greater) in our experiments. Regardless of the mechanism for this discrepancy, the previously reported autonomous chaperone function of this region of Ahsa1 may shed light on how Ahsa1 regulates Hsp90 function. That mutations in the NxNNWHW impair ATPase stimulation by full-length Ahsa1 to a greater degree than deletion of the entire ICD/NxNNWHW

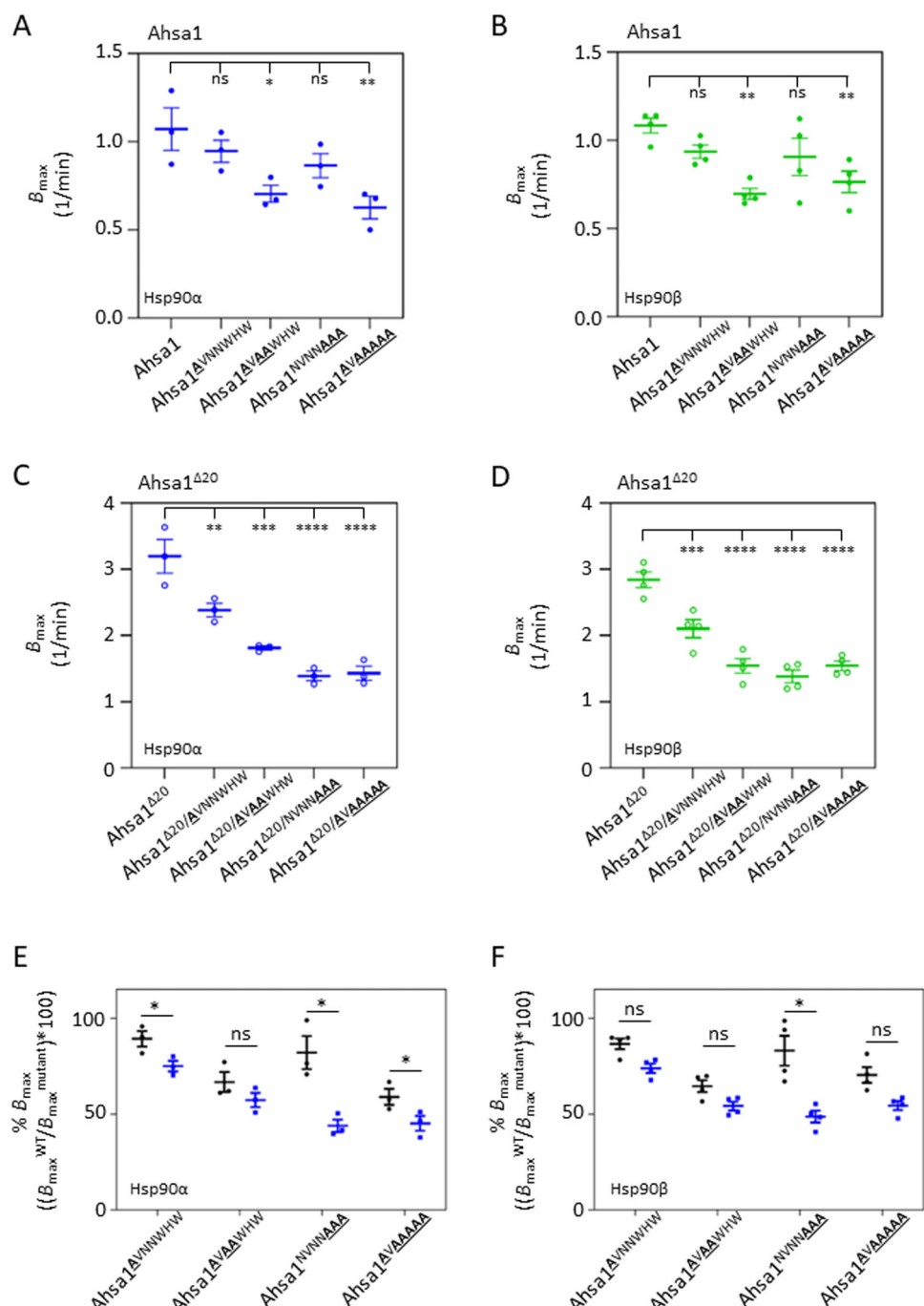

**Figure 4.  Mutations in the WHW portion of the NxNNWHW motif impair ATPase stimulation by Ahsa1$^{\Delta20}$ to a greater degree than full-length Ahsa1.**

$B_{max}$ values of reactions shown in Fig. 4. (A) $B_{max}$ values of point mutants of full-length Ahsa1 for Hsp90α (blue circles). (B) $B_{max}$ values of point mutants of full-length Ahsa1 for Hsp90β (green circles). (C) $B_{max}$ values of point mutants of Ahsa1$^{\Delta20}$ for Hsp90α (empty blue circles). (D) $B_{max}$ values of point mutants of Ahsa1$^{\Delta20}$ for Hsp90β (empty green circles). (E) $B_{max}$ values from (A, C) (Hsp90α) are expressed as a percent of the $B_{max}$ of their parental construct (Full-length Ahsa1 in black; Ahsa1$^{\Delta20}$ shown in blue). (F) $B_{max}$ values from (B, D) (Hsp90β) are expressed as a percent of the $B_{max}$ of their parental construct (Full-length Ahsa1 in black; Ahsa1$^{\Delta20}$ shown in blue). Data information: In (A–D), $B_{max}$ values from three independent experiments are plotted as well as the mean $+/-$ SEM ($N = 3$; each $N$ is one experiment carried out with technical triplicates as described in "Methods"). Statistical significance was calculated for (A–D) using a Dunnett's multiple comparison's test (*$P \leq 0.05$; **$P \leq 0.01$; ***$P \leq 0.001$; ****$P \leq 0.0001$). (E, F) Statistical significance was calculated between the $B_{max}$ of full-length Ahsa1 and Ahsa1$^{\Delta20}$ with Hsp90α (data from A, C) or Hsp90β (data from B, D), respectively, using an unpaired $t$ test (*$P \leq 0.05$; **$P \leq 0.01$; ***$P \leq 0.001$; ****$P \leq 0.0001$). $N = 3$. Source data are available online for this figure.

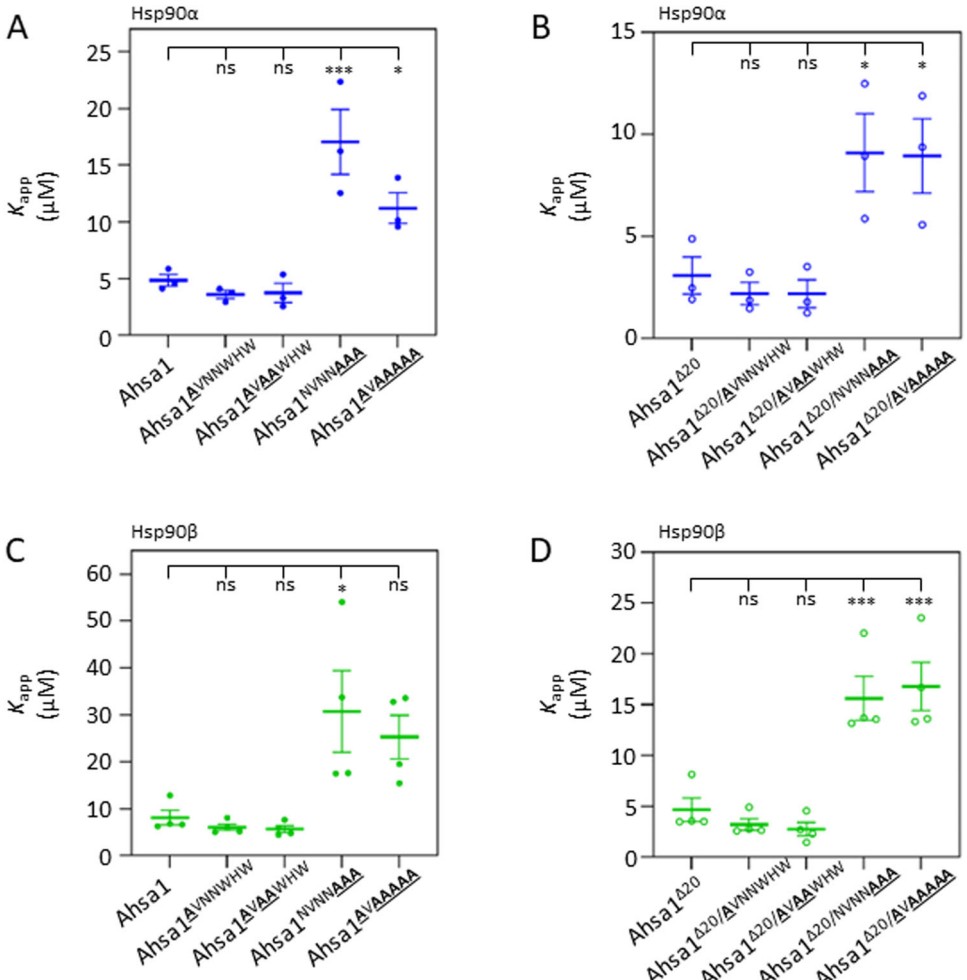

**Figure 5. Mutations in the WHW portion of the NxNNWHW motif reduce apparent affinity for Hsp90.**

$K_{app}$ values of reactions shown in Fig. 4. (A) Apparent affinity of point mutants of full-length Ahsa1 for Hsp90α (blue circles). (B) Apparent affinity of point mutants of Ahsa1$^{\Delta20}$ for Hsp90α (empty blue circles). (C) Apparent affinity of point mutants of full-length Ahsa1 for Hsp90β (green circles). (D). Apparent affinity of point mutants of Ahsa1$^{\Delta20}$ for Hsp90β (empty green circles). Data Information: In (A, B), $K_{app}$ values from three independent experiments are plotted as well as the mean $+/-$ SEM ($N = 3$; each N is one experiment carried out with technical triplicates as described in "Methods"). (C, D) $K_{app}$ values from four independent experiments are plotted as well as the mean $+/-$ SEM ($N = 4$; each N is one experiment carried out with technical triplicates as described in "Methods"). Statistical significance was calculated using a Dunnett's multiple comparisons test (*$P \le 0.05$; ***$P \le 0.001$). Source data are available online for this figure.

(in Ahsa1$^{\Delta27}$) illustrates the fact that the ICD inhibits more than one facet of this process. If one assumes that the structure of the yeast Hsp90/Aha1 complex (preprint: Liu et al, 2020) is similar to that of the mammalian proteins, it is likely the ICD is located very near the Hsp90 N domains and could potentially interfere with N terminal dimerization, ATP lid dynamics, or strand exchange. Any of these inhibitory mechanisms would not be associated with Ahsa1$^{\Delta27}$ which may explain its higher stimulation rate than some of the full-length NxNNWHW mutants.

Ahsa1 plays a role in the fate of important Hsp90 clients such as Tau and CFTR (Shelton et al, 2017; Wang et al, 2006). In the case of CFTR, cellular Ahsa1 levels appear to govern the balance between folding and degradation (Wang et al, 2006). This model is consistent with other chaperone cycles where attempts to fold will be made for a certain period of time or number of cycles (i.e. in the case of the calnexin/calreticulin cycle) (Ruddock and Molinari,

2006). In the case of Hsp90 and Ahsa1, Ahsa1 could be recruited to switch the function of Hsp90 from folding to degradation. The previously identified ICD (that can interact with misfolded proteins) could provide the means for Ahsa1 to recognize Hsp90 complexes with persistently misfolded clients. Alternatively, Ahsa1 could be capable of recognizing clients for delivery to Hsp90. However, our co-IP data with Ahsa1$^{E67K}$ (that cannot bind Hsp90) shows no interaction with GR suggesting that this is not the case for this client. That we did not recover Hsp90 or GR in complex with Ahsa1$^{\Delta20}$ suggests two possible scenarios for the effect of deletion of the ICD (Fig. 9B). Loss of the ICD may prevent recruitment to the Hsp90/GR complex at all but another possibility is that the absence of the ICD results in such rapid ATP hydrolysis that complexes with Hsp90 and GR cannot be recovered. More work will be required to determine the mechanics and functional consequences of Ahsa1 interaction with Hsp90.

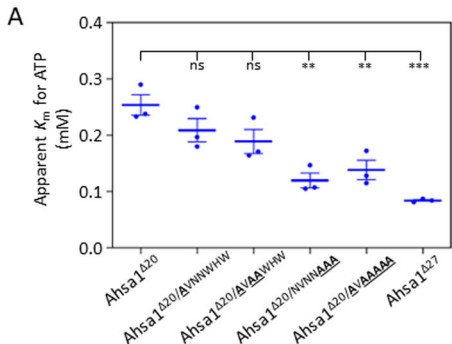

**Figure 6.   Mutations in the WHW portion of the NxNNWHW motif reduce the apparent $K_m$ of Hsp90 for ATP.**

Mutations in the WHW portion of the NxNNWHW motif results in higher apparent affinity for ATP in both Hsp90α (**A**) and Hsp90β (**B**). The apparent $K_m$ values for three experiments are shown in the scatter plots. Data Information: In (**A**, **B**), the apparent $K_m$ values for three independent experiments are plotted as well as the mean $+/-$ SEM ($N = 3$; each $N$ is one experiment carried out with technical triplicates as described in "Methods"). Statistical significance was calculated using a Dunnett's multiple comparisons test (*$P \leq 0.05$; **$P \leq 0.01$; ***$P \leq 0.001$). $N = 3$. Source data are available online for this figure.

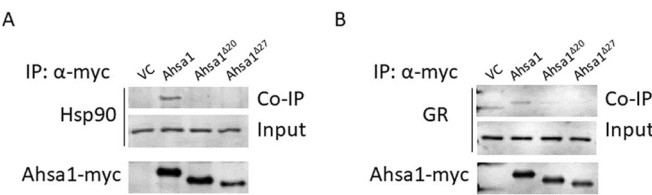

**Figure 7.   Loss of the ICD or ICD and NxNNWHW motif results in a loss of recruitment to Hsp90/client complexes.**

MDA-MB-231 cells were transfected with pcDNA5/TO plasmids encoding full-length Ahsa1, Ahsa1$^{\Delta 20}$, or Ahsa1$^{\Delta 27}$, each with a C-terminal myc tag. Cells were lysed and Ahsa1 was immunoprecipitated using anti-myc magnetic beads. Total cell lysates and immunoprecipitated material were analyzed with antibodies to Hsp90 and total Ahsa1 or the glucocorticoid hormone receptor and anti-myc antibodies. (**A**) Hsp90 and (**B**) glucocorticoid receptor is recovered in complex with full-length Ahsa1 but not Ahsa1$^{\Delta 20}$ or Ahsa1$^{\Delta 27}$ (**A**: $N = 5$; **B**: $N = 4$). Source data are available online for this figure.

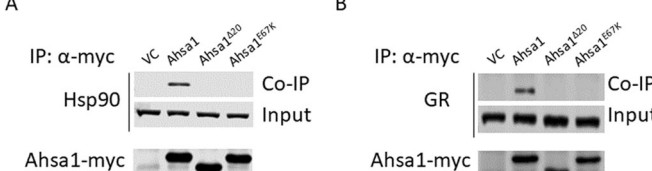

**Figure 8.   Complex formation between Ahsa1 and GR requires interaction between Ahsa1 and Hsp90.**

MDA-MB-231 cells were transfected with pcDNA5/TO plasmids encoding full-length Ahsa1, Ahsa1$^{\Delta 20}$, or Ahsa1$^{E67K}$, each with a C-terminal myc tag. Cells were lysed and Ahsa1 was immunoprecipitated using anti-myc magnetic beads. Total cell lysates and immunoprecipitated material were analyzed with antibodies to Hsp90 and total Ahsa1 or the glucocorticoid hormone receptor and anti-myc antibodies. (**A**) Hsp90 and (**B**) glucocorticoid receptor is recovered in complex with full-length Ahsa1 but not Ahsa1$^{\Delta 20}$ or Ahsa1$^{E67K}$ ($N = 3$). Source data are available online for this figure.

Our work here reveals several similarities between Ahsa1 and Aha1. The NxNNWHW motif appears to play a similar role in ATPase stimulation in both the human and yeast systems. If we consider Ahsa1$^{\Delta 20}$ to be more like yeast Aha1 in that it is missing the ICD, the deletion of the NxNNWHW motif from either of these constructs results in a dramatic decrease in ATPase stimulation (Mercier et al, 2019). Similarly, just as yeast Aha1$^{\Delta 11}$ (lacking the NxNNWHW motif) results in a much lower apparent $K_m$ for ATP than Hsp90 in complex with Aha1 (Mercier et al, 2019), the apparent $K_m$ for ATP of either Hsp90α or Hsp90β is lower with Ahsa1$^{\Delta 27}$ than with Ahsa1$^{\Delta 20}$.

Our data suggest that the NxNNWHW may have separable functions. Point mutations in the NxNNWHW motif resulted in variable degrees of impairment in ATPase stimulation capability in both the full-length Ahsa1 as well as Ahsa1$^{\Delta 20}$. Interestingly, mutations in the WHW portion of the NxNNWHW motif disproportionately affected the Ahsa1$^{\Delta 20}$ construct compared to full-length Ahsa1. Moreover, mutations in the WHW portion of the NxNNWHW motif dramatically affected both the apparent affinity of Ahsa1 for Hsp90 and the apparent $K_m$ of Hsp90 (α or β) for ATP. This suggests that the NxNN and WHW portions play

different roles in Hsp90 ATPase stimulation. This is consistent with the location of these two regions of the NxNNWHW motif in recently solved structures of the yeast Aha1-Hsp90 complex (preprint:Liu et al, 2020). These structures show that the WHW portion of the NxNNWHW motif is oriented towards Hsp90, while the NxNN residues make intramolecular contact with other parts of Aha1 (Fig. 9C). This may explain why mutations in the WHW portion of the motif affected the apparent affinity of Ahsa1 for Hsp90 as well as the apparent affinity for Hsp90 to ATP, but mutations in the NxNN did not. Further work will be required to understand the structural dynamics at the interface between Ahsa1 and Hsp90 at this site and the impact on ATPase activity.

# Methods

## Plasmids

pET11d bacterial expression vectors encoding C-terminally His-tagged Ahsa1 wildtype, Ahsa1$^{\Delta 20}$ and Ahsa1$^{\Delta 27}$ with and without NxNNWHW point mutants were obtained by Precision Bio and

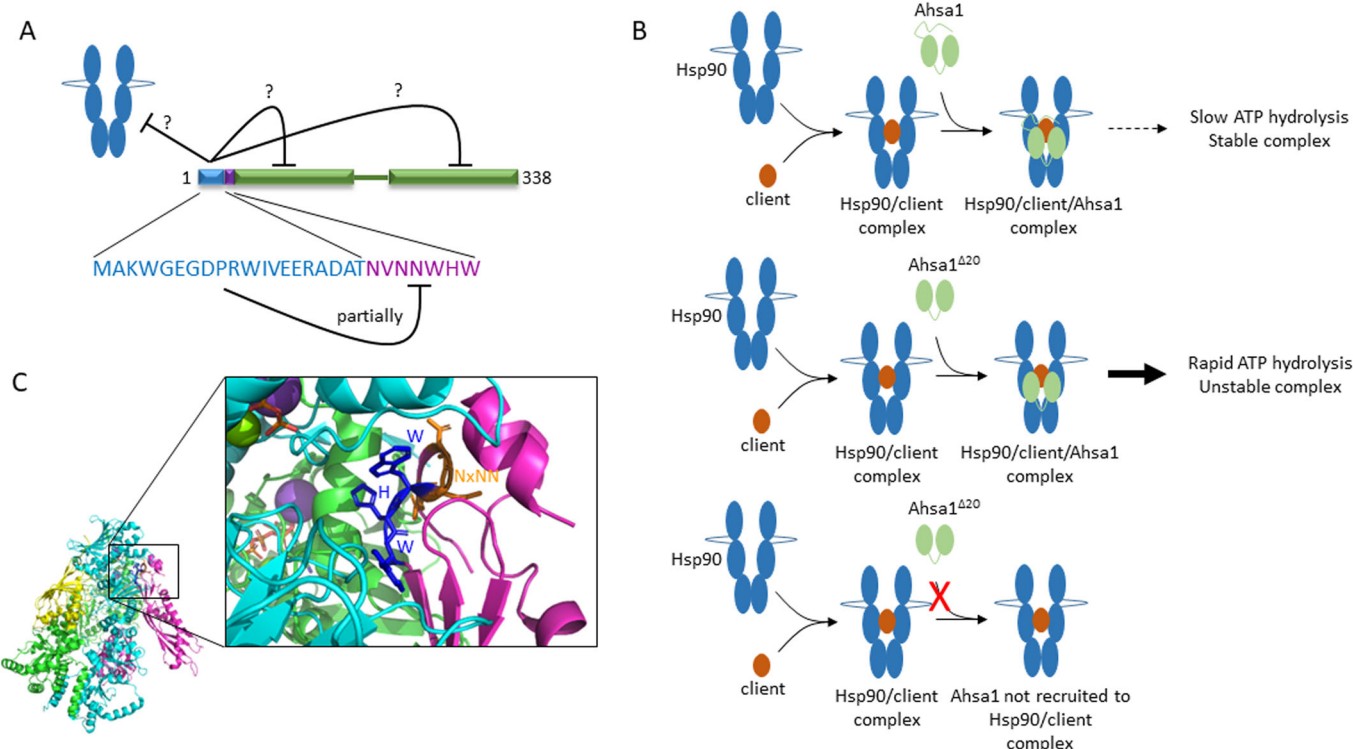

**Figure 9. Model of ICD autoinhibition of Ahsa1-stimulated Hsp90 ATPase activity, and Ahsa1 mediated client folding.**

(A) ICD only partially inhibits the NxNNWHW motif, whether it inhibits Hsp90 directly or another region of Ahsa1 is unknown. (B) Three possible roles for the ICD in complex recovery. (1) Ahsa1 is recruited in an ICD-dependent manner to Hsp90/GR complexes (TOP), and the complex remains stable because of slow ATP hydrolysis. (2) Ahsa1$^{\Delta 20}$ is recruited in an ICD-independent manner to Hsp90/GR complexes (MIDDLE) but rapid ATP hydrolysis results in destabilization of the complex. (3) Deletion of the ICD prevents Ahsa1 recruitment to Hsp90/GR complexes (BOTTOM). (C) Pymol structure (6XLF) showing the extensive contacts made by the yeast Aha1p (shown in pink) residues WHW (dark blue; residues 9, 10, and 11 in yeast Aha1p) with Hsp90 (cyan). NxNN residues are shown in orange.

verified by sanger sequencing. pcDNA5/TO mammalian expression vectors encoding C-terminally myc-tagged Ahsa1 wildtype, Ahsa1$^{\Delta 20}$ and Ahsa1$^{\Delta 27}$ was also obtained by Precision Bio and verified by Sanger sequencing. pET151 N-terminally His-tagged Hsp90α and Hsp90β were a kind gift from Dr. Sue-Ann Mok of the University of Alberta.

## Protein expression and purification

His-tagged Human Hsp90 and Ahsa1 constructs in pET151 and pET11d bacterial expression vectors, respectively, were transformed into *Escherichia coli* strain BL21 (DE3) (New England Biolabs) and plated on LB agar plates with 100 μg/ml ampicillin. Cells were grown in LB with 100 μg/ml ampicillin at 37 °C to an OD$_{600}$ of 0.8–1.0 and then induced with 0.5 mM isopropyl-1-thio-D-galactopyranoside (IPTG) for four hours. Cells were harvested by centrifugation and stored at −80 °C. Cells that expressed N-terminally His-tagged human Hsp90α or Hsp90β protein were resuspended in a lysis buffer containing 50 mM KH$_2$PO$_4$, pH 8.0, 500 mM KCl, 10 mM Imidazole pH 8.0, 10% Glycerol, 6 mM β-mercaptoethanol (βME), and protease inhibitor cocktail (Abcam). Cells containing C-terminally His-tagged Ahsa1 constructs were resuspended in a lysis buffer containing 25 mM NaH$_2$PO$_4$, pH 7.2, 500 mM NaCl, 1 mM MgCl$_2$, 20 mM Imidazole, 6 mM βME and protease inhibitor cocktail (Abcam). Cells were lysed using an Emulsiflex C3 (Avestin), and lysates were clarified by

ultracentrifugation. His-tagged proteins were purified by affinity chromatography on a HisTrap FF column using an AKTA Explorer FPLC (GE Healthcare). For Ahsa1, proteins were eluted with 25 mM NaH$_2$PO$_4$, pH 7.2, 500 mM NaCl, 1 mM MgCl$_2$, 1 M Imidazole and 6 mM βME. For Hsp90, proteins were first washed with an ATP-containing buffer (30 mM Tris pH 8.0, 500 mM KCl, 2 mM ATP, 5 mM MgCl$_2$, 0.1% Tween and 6 mM βME) and a high salt and low salt containing buffer (30 mM Tris pH 8.0, 500 mM KCl, 6 mM βME and 30 mM Tris pH 8.0, 50 mM KCl, 6 mM βME respectively) before being eluted in 30 mM Tris pH 8.0, 50 mM KCl, 6 mM βME and 250 mM Imidazole. Hsp90 was further purified by anion exchange using a Mono Q column or a HiTrap Q HP column with 30 mM Tris pH 8.0, 50 mM KCl, and 6 mM βME and eluted with a linear 0–50% gradient of 1 M KCl. Isolated proteins were then concentrated and further purified by size exclusion chromatography on a Superdex 200 using 30 mM Tris pH 7.5, 20 mM, KCl, 10% glycerol, and 1 mM DTT for Hsp90, and 25 mM HEPES pH 7.2, 50 mM NaCl and 1 mM DTT for Ahsa1. Proteins were verified by coomassie-stained SDS-PAGE analysis and then concentrated, snap frozen, and stored at −80 °C.

## ATPase assays

ATPase assays were done as previously described. Each assay was carried out in a 96-well plate with a 100 μl volume or a 384-well

plate with a 50 μl volume. The ATPase assays measured steady-state ATP hydrolysis using an ATP regenerating system coupled to NADH. For 90 min, the decreased absorbance of NADH at 340 nm was measured using a Biotek Synergy 4 or Biotek synergy H1 microplate reader. Then using Beer's law, this was converted to micromoles of ATP to determine the ATPase rate over time. The calculated ATPase rates were then expressed per μM of Hsp90 (1/min). The final concentration of each reaction includes 25 mM HEPES pH 7.2, 1 mM phosphoenol pyruvate (Sigma–Aldrich), 0.6 mM NADH, >3.75 units/mL pyruvate kinase (Sigma–Aldrich), >5.5 units/mL lactate dehydrogenase (Sigma–Aldrich), 5 mM MgCl$_2$, 1 mM DTT, and 2 mM ATP (for the cochaperone titration experiments only). All reactions also included identical experimental triplicates containing 100 μM NVP-AUY922 or DMSO. The ATPase activity of the NVP-AUY922 groups were subtracted from the unquenched reactions in the DMSO group to correct for contaminating ATPase activity.

In the cochaperone titration experiments (Figs. 2 and 4), ATPase reactions were carried out with 2 μM of Hsp90α or Hsp90β and increasing concentration of indicated hAhsa1 constructs (2, 4, 8, 16, 32 μM). Fit lines were calculated according to the following equation ($Y = ((B_{max} * X)/(K_{app} + X)) + X_0$), and the $B_{max}$ and $K_{app}$ were both determined using GraphPad Prism. In the ATP titration experiments (Figs. 3 and 5), ATPase reactions were carried out with 4 μM of Hsp90α or Hsp90β, 40 μM of the indicated Ahsa1 construct, and increasing concentrations of ATP (12.5, 25, 50, 100, 200, 400, 800, 1600 μM). ATPase rates were analyzed with the Michaelis–Menten nonlinear regression function in GraphPad Prism (Curve fits all had $R^2$ values greater than 0.9.), and the apparent $K_m$ for ATP was obtained.

### ITC experiments

All ITC measurements were performed at 25 °C with a stirring speed of 750 rpm in high gain mode and reference power of 5 μcal/s to determine the $K_D$, stoichiometry, and thermodynamic parameters of ligand binding using MicroCal PEAQ-ITC (Malvern Instruments Ltd., USA) calorimeter. Proteins and ligands were dialyzed against interaction buffer (30 mM Tris pH 7.5, 20 mM, KCl, 10% glycerol, and 1 mM Tris(2-carboxyethyl)phosphine hydrochloride). Three independent experiments were performed. Ligands in the syringe (Ahsa1, Ahsa1$^{\Delta 20}$) at a concentration of 200–300 μM were injected in the cell containing Hsp90-α at a concentration of 30 μM. The reference cell was filled with water. The first "technical" injection of a reduced volume (0.4 μl) was discarded during the analysis and was followed by 19 injections of 2 μl. Data were fitted to a two-site-binding model using the Malvern Analysis software. The automatically adjusted integration regions were used to minimize the impact of the researcher's arbitrary decision. The binding isotherms were fit to determine apparent molar reaction enthalpy (ΔH), apparent entropy (ΔS), dissociation constant ($K_D$), and stoichiometry of binding (N). The two-site-binding model was used to describe the isotherm and estimate binding parameters.

### Mammalian cell culture, transfection, immunoprecipitation, and immunoblotting

MDA-MB-231s were obtained from ATCC (human breast adenocarcinoma; ATCC HTB-26) and were cultured in Roswell Park Memorial Institute 1640 Medium (RPMI— Gibco)

supplemented with 10% fetal bovine serum (FBS—Corning). MDA-MB-231s were transiently transfected with pcDNA5/TO mammalian expression vectors encoding Ahsa1-myc-tagged constructs using the Lipofectamine 2000 Transfection Reagent (Invitrogen). One to three days after transfection, cells were washed with PBS twice and then lysed on ice with a lysis buffer containing 50 mM HEPES pH 7.4, 100 mM potassium acetate, 5 mM MgCl$_2$, 20 mM sodium molybdate, 2 mM orthovanadate, 20% glycerol, 1% TX-100, 20 mM N-ethylmaleimide, PhosSTOP (Roche), and protease inhibitor cocktail (Abcam). Cells were then scraped using a cell scraper and spun at 13,000 rpm for 20 min at 4 °C. Cell lysates were incubated with anti-c-Myc magnetic beads (Pierce; 88843) for one to three days at 4 °C. Beads were washed once with lysis buffer and then eluted in 5× laemmli buffer. Proteins were run on SDS-Page and western blot for analysis. Ahsa1 was detected by mouse anti-Ahsa1 5D11 monoclonal antibody (c/o Dr. William Balch, The Scripps Research Institute) or anti-myc 9E10 monoclonal antibody (Evan et al, 1985). Hsp90 was detected by rabbit anti-Hsp90 polyclonal antibody (Enzo; ADI-SPA-846-F). Glucocorticoid receptor was detected by rabbit anti-glucocorticoid receptor monoclonal antibody (Cell Signaling Technology; D6H2L).

### Densitometry

Band intensity was quantified using Image Studio™ software. Background binding of Hsp90 or GR (from the vector control lane) was subtracted from each co-IP. The resultant intensity was then normalized to the amount of Ahsa1 recovered in the IP and expressed as a percentage of the Hsp90 or GR recovered in complex with wildtype Ahsa1.

## Data availability

This study contains no data deposited in external repositories.

The source data of this paper are collected in the following database record: biostudies:S-SCDT-10_1038-S44319-024-00193-8.

## Peer review information

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

## Acknowledgements

The authors would like to thank Dr. Sue-Ann Mok for helpful advice on the purification of human Hsp90α and Hsp90β. This work was supported with funding from the Canadian Institutes of Health Research (178282) and the Natural Sciences and Engineering Research Council (RGPIN-2019-04967, PLP and 2018-04994, MO) and the Canada Foundation for Innovation (38496).

## Author contributions

**Solomon K Hussein**: Conceptualization; Data curation; Formal analysis; Validation; Investigation; Visualization; Methodology; Writing—review and editing. **Rakesh Bhat**: Conceptualization; Formal analysis; Validation; Investigation; Visualization; Methodology; Writing—review and editing. **Michael Overduin**: Resources; Supervision; Funding acquisition; Methodology; Writing—review and editing. **Paul LaPointe**: Conceptualization; Resources; Data curation; Formal analysis; Supervision; Funding acquisition; Investigation; Visualization; Methodology; Writing—original draft; Project administration; Writing—review and editing.

Source data underlying figure panels in this paper may have individual authorship assigned. Where available, figure panel/source data authorship is listed in the following database record: biostudies:S-SCDT-10_1038-S44319-024-00193-8.

## Disclosure and competing interests statement

The authors declare no competing interests.

# Expanded View Figures

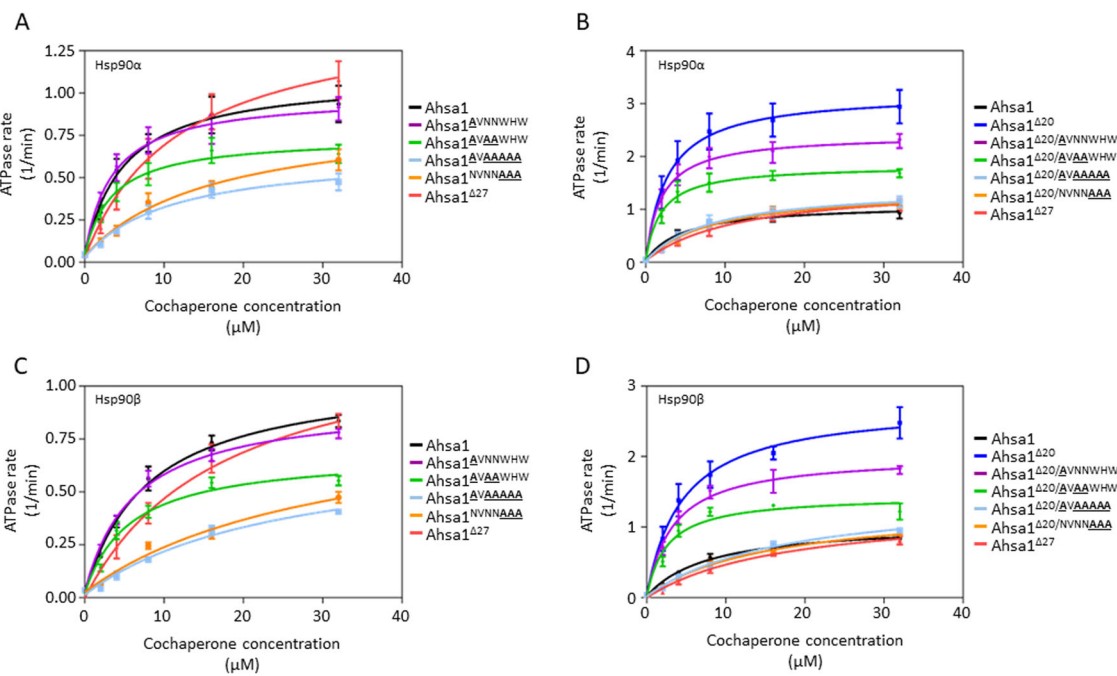

**Figure EV1. Point mutations in the NxNNWHW motif impair ATPase stimulation by Ahsa1.**

(A) Stimulation of Hsp90α ATPase activity by increasing concentrations of full-length Ahsa1 (black), Ahsa1$^{AVNNWHW}$ (purple), Ahsa1$^{AVAAWHW}$ (green), Ahsa1$^{NVNNAAA}$ (orange), Ahsa1$^{AVAAAAA}$ (light blue), and Ahsa1$^{Δ27}$ (red). (B) Stimulation of Hsp90α ATPase activity by increasing concentrations of full-length Ahsa1 (black), Ahsa1$^{Δ20}$ (blue), Ahsa1$^{Δ20/AVNNWHW}$ (purple), Ahsa1$^{Δ20/AVAAWHW}$ (green), Ahsa1$^{Δ20/NVNNAAA}$ (orange), Ahsa1$^{Δ20/AVAAAAA}$ (light blue), and Ahsa1$^{Δ27}$ (red). (C) Stimulation of Hsp90β ATPase activity by increasing concentrations of full-length Ahsa1 (black), Ahsa1$^{AVNNWHW}$ (purple), Ahsa1$^{AVAAWHW}$ (green), Ahsa1$^{NVNNAAA}$ (orange), Ahsa1$^{AVAAAAA}$ (light blue), and Ahsa1$^{Δ27}$ (red). (D) Stimulation of Hsp90β ATPase activity by increasing concentrations of full-length Ahsa1 (black), Ahsa1$^{Δ20}$ (blue), Ahsa1$^{Δ20/AVNNWHW}$ (purple), Ahsa1$^{Δ20/AVAAWHW}$ (green), Ahsa1$^{Δ20/NVNNAAA}$ (orange), Ahsa1$^{Δ20/AVAAAAA}$ (light blue), and Ahsa1$^{Δ27}$ (red). Reactions contained 1 μM Hsp90α or Hsp90β and indicated concentration of cochaperone. Data Information: All data shown in (A, B) were obtained in the same 384-well experiment three times ($N = 3$). Data for full-length Ahsa1 constructs were plotted in (A) and for Ahsa1$^{Δ20}$ constructs were plotted in (B). In (A, B), data for each concentration of cochaperone are presented as mean $+/-$ SEM of 3 independent experiments ($N = 3$; each $N$ is one experiment carried out with technical triplicates as described in "Methods"). All data shown in (C, D) were obtained in the same 384-well experiment four times ($N = 4$). Data for full-length Ahsa1 constructs were plotted in C and for Ahsa1$^{Δ20}$ constructs were plotted in (D). (C, D) Data for each concentration of cochaperone are presented as mean $+/-$ SEM of 4 independent experiments ($N = 4$; each $N$ is one experiment carried out with technical triplicates as described in "Methods"). Source data are available online for this figure.

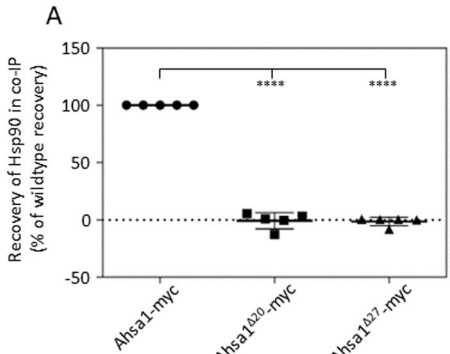
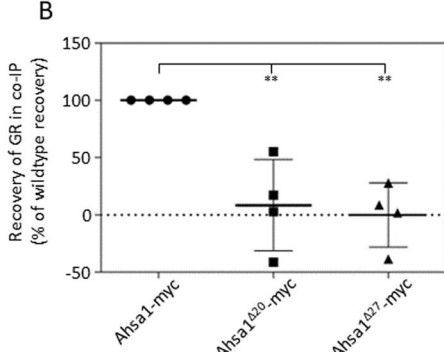

**Figure EV2. The ICD is required for stable interaction with Hsp90 and GR.**

Band intensity of five replicate experiments was measured by densitometry. After background subtraction, intensities of Hsp90 (**A**) and GR (**B**) were normalized against the amount of Ahsa1 recovered and expressed as a percentage of each protein with wildtype Ahsa1. Data Information: In (**A**), data from five individual experiments are plotted as individual points as well as the mean $+/-$ SD ($N = 5$). In (**B**), data from four individual experiments are plotted as individual points as well as the mean $+/-$ SD ($N = 4$). Statistical significance was calculated for (**A**, **B**) using a Dunnett's multiple comparison's test (**$P \leq 0.01$; ****$P \leq 0.0001$). Source data are available online for this figure.

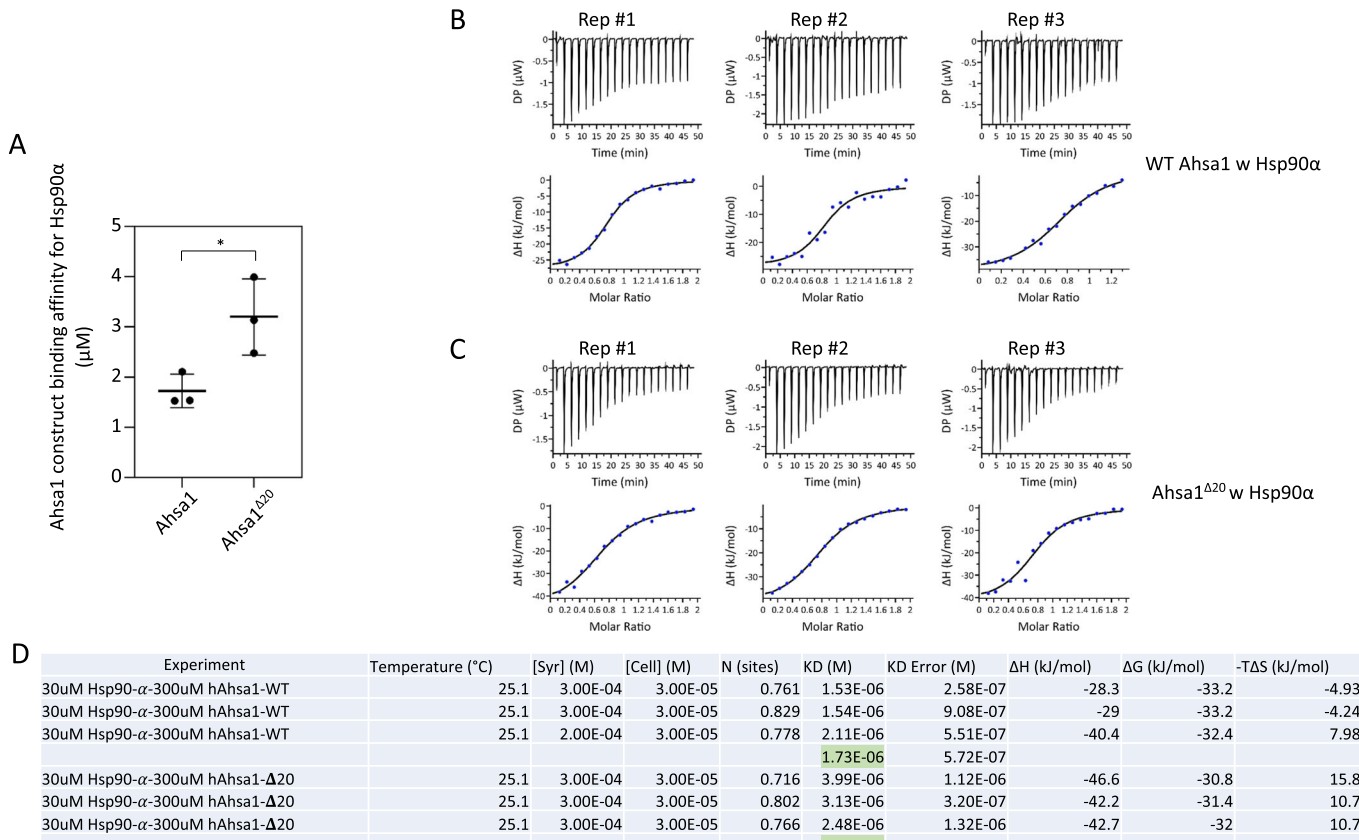

**D**

| Experiment | Temperature (°C) | [Syr] (M) | [Cell] (M) | N (sites) | KD (M) | KD Error (M) | ΔH (kJ/mol) | ΔG (kJ/mol) | -TΔS (kJ/mol) |
|---|---|---|---|---|---|---|---|---|---|
| 30uM Hsp90-α-300uM hAhsa1-WT | 25.1 | 3.00E-04 | 3.00E-05 | 0.761 | 1.53E-06 | 2.58E-07 | -28.3 | -33.2 | -4.93 |
| 30uM Hsp90-α-300uM hAhsa1-WT | 25.1 | 3.00E-04 | 3.00E-05 | 0.829 | 1.54E-06 | 9.08E-07 | -29 | -33.2 | -4.24 |
| 30uM Hsp90-α-300uM hAhsa1-WT | 25.1 | 2.00E-04 | 3.00E-05 | 0.778 | 2.11E-06 | 5.51E-07 | -40.4 | -32.4 | 7.98 |
|  |  |  |  |  | 1.73E-06 | 5.72E-07 |  |  |  |
| 30uM Hsp90-α-300uM hAhsa1-Δ20 | 25.1 | 3.00E-04 | 3.00E-05 | 0.716 | 3.99E-06 | 1.12E-06 | -46.6 | -30.8 | 15.8 |
| 30uM Hsp90-α-300uM hAhsa1-Δ20 | 25.1 | 3.00E-04 | 3.00E-05 | 0.802 | 3.13E-06 | 3.20E-07 | -42.2 | -31.4 | 10.7 |
| 30uM Hsp90-α-300uM hAhsa1-Δ20 | 25.1 | 3.00E-04 | 3.00E-05 | 0.766 | 2.48E-06 | 1.32E-06 | -42.7 | -32 | 10.7 |
|  |  |  |  |  | 3.20E-06 | 9.20E-07 |  |  |  |

**Figure EV3. Affinity of Ahsa1 and Ahsa$^{\Delta20}$ for Hsp90α.**

(A) Binding affinity of wildtype Ahsa1 and Ahsa1$^{\Delta20}$ for Hsp90α are shown in the scatter plot. $N = 3$. (B, C) The final ITC figures are generated with MicroCal PEAQ-ITC analysis software. The upper panels show the baseline-subtracted, singular-value-decomposition (SVD)-corrected thermograms and the bottom panels show the binding isotherms with fitting curves of Hsp90-α titration by hAhsa1-WT (B) or Ahsa1$^{\Delta20}$ (C), respectively. (D) Details of the thermodynamic parameters: (i.e. *n* stoichiometric ratio of ligand to protein, $K_D$ dissociation constant; ΔH, ΔS and ΔG indicate changes in enthalpy, entropy, and Gibbs free energy, respectively). Data Information: In (A), calculated affinities for three experiments are shown as individual points as well as the mean $+/-$ SD ($N = 3$). Statistical significance was calculated using an unpaired *t* test (\**P* ≤ 0.05). Source data are available online for this figure.

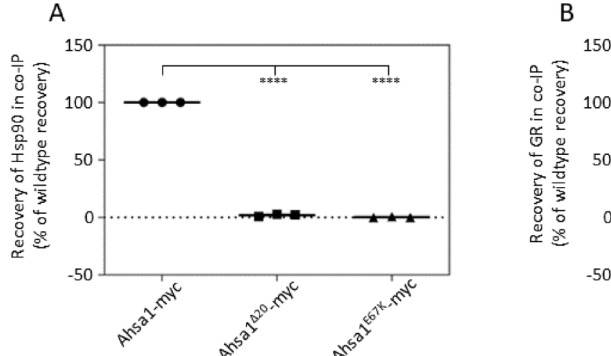

**Figure EV4. Complex formation between Ahsa1 and GR requires Ahsa1 interaction with Hsp90.**

Band intensity of three replicate experiments was measured by densitometry. After background subtraction, intensities of Hsp90 (**A**) and GR (**B**) were normalized against the amount of Ahsa1 recovered and expressed as a percentage of each protein with wildtype Ahsa1. Data Information: In (**A, B**), data from three individual experiments are plotted as individual points as well as the mean $+/-$ SD ($N = 3$). Statistical significance was calculated for (**A, B**) using a Dunnett's multiple comparison's test (***$P \leq 0.001$; ****$P \leq 0.0001$). Source data are available online for this figure.

