## [Peer Review File · EMBO Reports]

Recruitment of Ahsa1 to Hsp90 is regulated by a conserved peptide that inhibits ATPase stimulation

Solomon Hussein, Rakesh Bhat, Michael Overduin, and Paul LaPointe

Corresponding author(s): Paul LaPointe (paul.lapointe@ualberta.ca)

Review Timeline:

Submission Date:	15th Jun 23
Editorial Decision:	15th Aug 23
Revision Received:	21st Feb 24
Editorial Decision:	14th May 24
Revision Received:	23rd May 24
Editorial Decision:	11th Jun 24
Revision Received:	12th Jun 24
Accepted:	17th Jun 24

Editor: *Martina Rembold*

Transaction Report:

Dear Dr. LaPointe

Thank you for the submission of your research manuscript to our journal. I apologize for the delay in handling your manuscript but we have now received the full set of referee reports that is copied below.

As you will see, the referees acknowledge that the findings are potentially interesting, but they also raise a number of concerns (see below) that need to be addressed.

Given these constructive comments, we would like to invite you to revise your manuscript with the understanding that the referee concerns (as detailed above and in their reports) must be fully addressed and their suggestions taken on board. Please address all referee concerns in a complete point-by-point response. Acceptance of the manuscript will depend on a positive outcome of a second round of review. It is EMBO Reports policy to allow a single round of revision only and acceptance or rejection of the manuscript will therefore depend on the completeness of your responses included in the next, final version of the manuscript.

We realize that it is difficult to revise to a specific deadline. In the interest of protecting the conceptual advance provided by the work, we recommend a revision within 3 months (November 15th). Please discuss the revision progress ahead of this time with the editor if you require more time to complete the revisions.

I am also happy to discuss the revision further via e-mail or a video call, if you wish.

*****IMPORTANT NOTE:

We perform an initial quality control of all revised manuscripts before re-review. Your manuscript will FAIL this control and the handling will be DELAYED if the following APPLIES:

- 1) A data availability section providing access to data deposited in public databases is missing. If you have not deposited any data, please add a sentence to the data availability section that explains that.
- 2) Your manuscript contains statistics and error bars based on $n=2$. Please use scatter blots in these cases. No statistics should be calculated if $n=2$.

When submitting your revised manuscript, please carefully review the instructions that follow below. Failure to include requested items will delay the evaluation of your revision.*****

- 1) a .docx formatted version of the manuscript text (including legends for main figures, EV figures and tables). Please make sure that the changes are highlighted to be clearly visible.
- 2) individual production quality figure files as .eps, .tif, .jpg (one file per figure). Please download our Figure Preparation Guidelines (figure preparation pdf) from our Author Guidelines pages <https://www.embopress.org/page/journal/14693178/authorguide> for more info on how to prepare your figures.
- 3) a .docx formatted letter INCLUDING the reviewers' reports and your detailed point-by-point responses to their comments. As part of the EMBO Press transparent editorial process, the point-by-point response is part of the Review Process File (RPF), which will be published alongside your paper.
- 4) a complete author checklist, which you can download from our author guidelines (<<https://www.embopress.org/page/journal/14693178/authorguide>>). Please insert information in the checklist that is also reflected in the manuscript. The completed author checklist will also be part of the RPF.
- 5) Please note that all corresponding authors are required to supply an ORCID ID for their name upon submission of a revised manuscript (<<https://orcid.org/>>). Please find instructions on how to link your ORCID ID to your account in our manuscript tracking system in our Author guidelines (<<https://www.embopress.org/page/journal/14693178/authorguide#authorshipguidelines>>)
- 6) We replaced Supplementary Information with Expanded View (EV) Figures and Tables that are collapsible/expandable online. A maximum of 5 EV Figures can be typeset. EV Figures should be cited as 'Figure EV1, Figure EV2' etc... in the text and their respective legends should be included in the main text after the legends of regular figures.

7) Please note that a Data Availability section at the end of Materials and Methods is now mandatory. In case you have no data that requires deposition in a public database, please state so instead of refereeing to the database. See also <<https://www.embopress.org/page/journal/14693178/authorguide#dataavailability>>. Please note that the Data Availability Section is restricted to new primary data that are part of this study.

Additional information on source data and instruction on how to label the files are available <<https://www.embopress.org/page/journal/14693178/authorguide#sourcedata>>.

10) Figure legends and data quantification:
The following points must be specified in each figure legend:

- the name of the statistical test used to generate error bars and P values,
 - the number (n) of independent experiments (please specify technical or biological replicates) underlying each data point,
 - the nature of the bars and error bars (s.d., s.e.m.)
- If the data are obtained from n {less than or equal to} 5, show the individual data points in addition to the SD or SEM.
- If the data are obtained from n {less than or equal to} 2, use scatter blots showing the individual data points.

See also the guidelines for figure legend preparation:
<https://www.embopress.org/page/journal/14693178/authorguide#figureformat>

11) All Materials and Methods need to be described in the main text. We would encourage you to use 'Structured Methods', our new Materials and Methods format. According to this format, the Materials and Methods section should include a Reagents and Tools Table (listing key reagents, experimental models, software and relevant equipment and including their sources and relevant identifiers) followed by a Methods and Protocols section in which we encourage the authors to describe their methods using a step-by-step protocol format with bullet points, to facilitate the adoption of the methodologies across labs. More information on how to adhere to this format as well as downloadable templates (.doc or .xls) for the Reagents and Tools Table can be found in our author guidelines: <<https://www.embopress.org/page/journal/14693178/authorguide#manuscriptpreparation>>. An example of a Method paper with Structured Methods can be found here: <<https://www.embopress.org/doi/10.15252/msb.20178071>>.

12) Our journal encourages inclusion of *data citations in the reference list* to directly cite datasets that were re-used and obtained from public databases. Data citations in the article text are distinct from normal bibliographical citations and should directly link to the database records from which the data can be accessed. In the main text, data citations are formatted as follows: "Data ref: Smith et al, 2001" or "Data ref: NCBI Sequence Read Archive PRJNA342805, 2017". In the Reference list, data citations must be labeled with "[DATASET]". A data reference must provide the database name, accession number/identifiers and a resolvable link to the landing page from which the data can be accessed at the end of the reference. Further instructions are available at <<https://www.embopress.org/page/journal/14693178/authorguide#referencesformat>>.

13) As part of the EMBO publication's Transparent Editorial Process, EMBO Reports publishes online a Review Process File to accompany accepted manuscripts. This File will be published in conjunction with your paper and will include the referee reports,

your point-by-point response and all pertinent correspondence relating to the manuscript.

Yours sincerely,

Referee #1:

Firstly it was a pleasure to review such an informative and well written paper on Ahsa1 and Hsp90, which focuses on an important motif that can affect the ATPase "accelerator" of Hsp90.

I only have one significant comment on the paper, around figure 8: The legend needs greater explanation. As is currently only one sentence, and does not guide/inform the reader. The paper version I downloaded seems to have two different figure 8's present - one a full western (is this for supplemental) and a edited version too. If including both the first one needs the * (Ab chains?) identifying in the figure legend.

A minor comment, in the penultimate paragraph of the introduction, you refer to ICD of Ahsa1 as autoinhibits the Hsp90 stimulation activity. If I understand correctly, it does not as such fully inhibit but down regulates the affect? Maybe a small reword to clarify?

Otherwise, a rather good and informative piece of research that is very worthy of publication.

Referee #2:

The Hsp90 chaperone is regulated by a number of co-chaperones. Among them, Aha1 is known to bind to Hsp90 and to accelerate its ATPase activity. Mechanistically, the function of Aha1 is not understood and thus studies that shed light on this process are needed.

In this contribution, the authors identify a sequence motif at the N-terminus of Aha1, called ICD, which they show to have an auto-inhibitory effect on the ATPase stimulation of Hsp90. The authors suggest that the ICD interferes with the function of the adjacent NxNNWHW motif.

Overall, this is an interesting study, which provides important insight into a regulatory element of Aha1.

There are some open questions which need to be answered.

1. While the effects of the deletion of the ICD are novel and clear, the effects of the mutations of the NxNNWHW motif are less clear.

The authors assume that they two adjacent motifs influence each other directly. Specifically, the authors suggest a "masking" of the NxNNWHW by the ICD. The section on these experiments in the results is a bit confusing. The authors replace residues individually or together by alanines. They show that only replacing the three N together by A or the mutation of all N and W residues results in a reduced stimulatory activity of the Aha variant. This is not clearly stated in the results section. The deletion construct lacking the ICD is more sensitive to mutation. Here all mutations have an effect on the ATPase stimulation.

The conclusion by the authors is that masking occurs but that there also seems "some other mechanism". In the discussion, the authors refer to recent cryo-EM structures (without a reference) from the Agard lab concerning the position of specific parts of the NxNNWHW motif. They do not mention the ICD. Is this segment resolved in the structure? A more detailed discussion of the structure may be useful to explain the effects observed.

The individual panels of figure 5 are not referred to in the text in chronological order.

2. The model in figure 9 b focuses on client binding to Hsp90 in the presence of Aha1. The cartoon does not include the new information gained in this study nor the effect on closing of Hsp90 in general. Fig. 9c needs more labelling of the specific motifs depicted and maybe different colours.

3. Figures 2, 4, 5. The y-axis in the different graphs is different in scale and labelling. Please adjust to the same scale and digits.

4. page 3, second paragraph, "citation" should be replaced by the respective reference.
5. The number of figures could be reduced by combining panels of figs. 5-7. It seems not all analyses of the ATPase assays are needed.
6. Both Hsp90 and Aha constructs contain His-tags. Are there any experiments with tag-less versions?
7. There are two figures 8 a and b. The bands need to be quantified. I assume that the experiment was performed in triplicates. The legend to this figure needs to be improved. Some aspects of the results are in contrast to the in vitro data. One difference was the use of a Myc-tag here.

Referee #3:

The authors report the first 20 residues of the human Aha1 homologue to act as autoinhibitor. This sequence is conserved between metazoan but is not present in yeast. Given that Yeats Aha1 acted as paradigm to discover the function for Aha1, it had been overlooked so far. This is a very intriguing finding that is potentially important for regulation of Hsp90 chaperones. However, the findings are not sufficiently clearly presented to be convincing, raising some major concerns as outlines below.

Major concerns.

1. The authors claim that the N-terminal 20 residues act as autoinhibitor for Hsp90, suppress the ATPase and reduce affinity for Hsp90.
 - a. The authors need to determine the KiDs for AHSA1 binding to Hsp90, with and without the 20 N-terminal residues to substantiate this claim, via direct interaction measurements.
 - b. The authors need to clarify how the system should work - if the first 20 residues that act as autoinhibitor also reduce affinity for Hsp90, how is this break released? This needs to be experimentally supported to justify the central conclusion of the paper.
2. the authors report point mutant effects in the 21-27 rescue segment.
 - a. These mutants are less relevant for supporting the central conclusion than mutations in residues 1-20 would be. Thus, which residues in residues 1-20 are crucial?
 - b. Why is Delta27 in some experiments more effective than wild type? This suggests that residues 21-27 are not important for stimulating the ATPase of Hsp90, which seem to contradict central assumptions of the authors.
 - c. The mutations in residues 21-27 have rather mild effects, raising the question how important this region actually is. please clarify.
3. what would be the proposed mechanism for auto inhibition?
 - a. the authors provide a structural; model, but this does not clarify this point. Which are the interactions the authors think are key for the proposed effects?
 - b. Please annotate the relevant proteins Ana residues within the figure. At present the legend does not provide enough information to understand the figure.

Referee #1:

Firstly it was a pleasure to review such an informative and well written paper on Ahsa1 and Hsp90, which focuses on an important motif that can affect the ATPase "accelerator" of Hsp90. I only have one significant comment on the paper, around figure 8: The legend needs greater explanation. As is currently only one sentence, and does not guide/inform the reader. The paper version I downloaded seems to have two different figure 8's present -one a full western (is this for supplemental) and a edited version too. If including both the first one needs the * (Ab chains?) identifying in the figure legend.

This was an oversight on our part. We were anticipating having to include an uncropped version of the western blot and left it in the main figures by mistake. This has now been converted to a supplemental figure for review purposes. We have expanded the figure legend to better explain the experimental approach and how it was interpreted. We have also added more annotations in the uncropped supplemental version for the different bands observed.

A minor comment, in the penultimate paragraph of the introduction, you refer to ICD of Ahsa1 as autoinhibits the Hsp90 stimulation activity. If I understand correctly, it does not as such fully inhibit but down regulates the affect? Maybe a small reword to clarify? Otherwise, a rather good and informative piece of research that is very worthy of publication.

We agree that this word is problematic and too strongly suggests complete inhibition. We have clarified this in the text and have also amended the title to better capture what the data actually shows.

We would like to thank the reviewer for their positive thoughts on the work!

Referee #2:

The Hsp90 chaperone is regulated by a number of co-chaperones. Among them, Aha1 is known to bind to Hsp90 and to accelerate its ATPase activity. Mechanistically, the function of Aha1 is not understood and thus studies that shed light on this process are needed.

In this contribution, the authors identify a sequence motif at the N-terminus of Aha1, called ICD, which they show to have an auto-inhibitory effect on the ATPase stimulation of Hsp90. The authors suggest that the ICD interferes with the function of the adjacent NxNNWHW motif. Overall, this is an interesting study, which provides important insight into a regulatory element of Aha1.

There are some open questions which need to be answered.

1. While the effects of the deletion of the ICD are novel and clear, the effects of the mutations of the NxNNWHW motif are less clear.

The authors assume that they two adjacent motifs influence each other directly. Specifically, the authors suggest a "masking" of the NxNNWHW by the ICD. The section on these experiments in the results is a bit confusing. The authors replace residues individually or together by alanines. They show that only replacing the three N together by A or the mutation of all N and W residues results in a reduced stimulatory activity of the Aha variant. This is not clearly stated in the results section. The deletion construct lacking the ICD is more sensitive to mutation. Here all mutations have an effect on the ATPase stimulation.

We have tried to clarify the rationale for both the experiments as well as how we interpret them. The striking increase in ATPase stimulation we observed with Ahsa1^{Δ20} left us with obvious questions for the mechanism. We reasoned that, since the 20 amino acid ICD is immediately next to the NxNNWHW motif (which we had previously shown with yeast proteins to be important for stimulation (Mercier et. al. Nat Comms 2019)) we thought the simplest (and somewhat wishful) explanation would be that the ICD was simply masking the NxNNWHW. If this were the case, we would expect that mutation in residues of the NxNNWHW would greatly diminish stimulation in Ahsa1^{Δ20} but have no effect in the context of full length Ahsa1. We did not get such a clear-cut result in these experiments. Rather, the mutations in the NxNNWHW motif affected ATPase stimulation by both full length Ahsa1 and Ahsa1^{Δ20}, but these defect were more severe in the context of Ahsa1^{Δ20}. We still interpret this to mean that the ICD in some way interferes with the NxNNWHW motif, either directly or indirectly, but that there is also another way that the ICD diminishes ATPase stimulation by Ahsa1. Given its proximity to the N domains of Hsp90, and the immense importance of the rearrangements that must occur here for ATPase activity to occur, there are many possibilities. We (again, somewhat wishfully) attempted to measure the effects of a synthetic peptide corresponding to the ICD (and an inverted control peptide) in our ATPase assays. The addition of these peptides did not alter ATPase rates with any of the Ahsa1 constructs so we could not draw any conclusions (since we had no positive control for the peptide behaving as it 'should' in the first place).

The conclusion by the authors is that masking occurs but that there also seems "some other mechanism". In the discussion, the authors refer to recent cryo-EM structures (without a reference) from the Agard lab concerning the position of specific parts of the NxNNWHW motif. They do not mention the ICD. Is this segment resolved in the structure? A more detailed discussion of the structure may be useful to explain the effects observed.

We have now added this reference – this was an oversight on our part (and we were also unsure of the policies regarding the citation of papers from BioRxiv). We thank the reviewer for highlighting a major question here. The structures in question are with the *yeast* Hsp90 and Aha1 proteins. The so-called ICD is not present in the yeast Aha1 so there is nothing to be learned from the structures about the precise position of this segment of the mammalian Ahsa1. However, one can infer speculatively, if the structure of the mammalian complex is similar, that the ICD would be oriented towards or near the dimerization interface of the Hsp90 N domains. One can imagine many ways that the ICD could interfere with the acquisition of structure of the adjacent NxNNWHW motif in Aha1 or with the strap or dimerization helix in the Hsp90 N domains.

The individual panels of figure 5 are not referred to in the text in chronological order.

Thank you for pointing this out. We have reformatted the figure to a different layout. The figure now matches the order in the text.

2. The model in figure 9 b focuses on client binding to Hsp90 in the presence of Aha1. The cartoon does not include the new information gained in this study nor the effect on closing of Hsp90 in general. Fig. 9c needs more labelling of the specific motifs depicted and maybe different colours.

The new data we added with the Ahsa1^{E67K} mutant allows us to discard one of the potential scenarios in this figure. We have remade it and have tried to better illustrate the role of the ICD.

3. Figures 2, 4, 5. The y-axis in the different graphs is different in scale and labelling. Please adjust to the same scale and digits.

The scale of each graph was chosen to reflect the range of activities being plotted. Hsp90 α had a higher stimulated rate than Hsp90 β so they were plotted to capture the range of rates for that experiment. Replotting the Hsp90 beta curves on the scale of the Hsp90 alpha would leave a large amount of blank space at the top of the graph. For the sake of clarity and aesthetics, we scaled within an individual experiment. It should be noted that comparing the rates in 4B to 4D wasn't our intention. However, we did not use consistent digits in each figure and this has now been corrected for consistency. We have decided to move the whole of figure 4 to supplemental because it is largely redundant with the plots of Bmax in Figure 5.

4. page 3, second paragraph, "citation" should be replaced by the respective reference.

Thank you for pointing this out. The appropriate citation has been properly added.

5. The number of figures could be reduced by combining panels of figs. 5-7. It seems not all analyses of the ATPase assays are needed.

We have made Figure 4 a supplemental figure but feel figures 5 through 7 are all reporting on a different enzymatic parameter. Figure 5 shows the Bmax rates, figure 6 shows the apparent affinity of the different co-chaperone constructs, and figure 7 shows the apparent affinity for ATP. We feel the data is better left separate given the large number of panels that would be present in a combined figure. We hope the reviewer will agree.

6. Both Hsp90 and Aha constructs contain His-tags. Are there any experiments with tag-less versions?

This is correct. All of our recombinant proteins have His tags (N terminal for Hsp90; C terminal for Aha1). We have not conducted experiments with tag-less versions but it is important to note that all of the Ahsa1 constructs have the same tag in the same location so the differences we see with our truncations and point mutations are unlikely to be attributable to the tag.

7. There are two figures 8 a and b. The bands need to be quantified. I assume that the experiment was performed in triplicates. The legend to this figure needs to be improved. Some aspects of the results are in contrast to the in vitro data. One difference was the use of a Myc-tag here.

This was an oversight on our part. We were anticipating having to include an uncropped version of the western blot and left it in the main figures by mistake. This has now been removed. We have included a quantification of the band intensities in the supplemental figures.

Referee #3:

The authors report the first 20 residues of the human Aha1 homologue to act as autoinhibitor. This sequence is conserved between metazoan but is not present in yeast. Given that Yeats Aha1 acted as paradigm to discover the function for Aha1, it had been overlooked so far. This is a very intriguing finding that is potentially important for regulation of Hsp90 chaperones. However, the findings are not sufficiently clearly presented to be convincing, raising some major concerns as outlines below.

Major concerns.

1. The authors claim that the N-terminal 20 residues act as autoinhibitor for Hsp90, suppress the ATPase and reduce affinity for Hsp90.

Just to clarify this –

a. The authors need to determine the K_i Ds for AHSA1 binding to Hsp90, with and without the 20 N-terminal residues to substantiate this claim, via direct interaction measurements.

The apparent affinity we measured in our ATPase assays only describes the interaction that results in stimulation. However, since there are two binding sites for Ahsa1 on every Hsp90 dimer, we can't comment on the affinity for the second site since it doesn't result in additional stimulation. We considered using SPR to address this concern but my work as a postdoc made me reluctant because the chip coupling can interfere with conformational changes in Hsp90 (or Aha1) that occur during the rather complex binding process. Instead, we did ITC and, consistent with what we inferred from our ATPase data, we did not observe a large difference between Ahsa1 and Ahsa1^{Δ20} with respect to their affinity for Hsp90. The binding affinities also matched quite well with the apparent affinities we measured using the ATPase assay. We have included this new data as a supplemental figure (S3).

b. The authors need to clarify how the system should work - if the first 20 residues that act as autoinhibitor also reduce affinity for Hsp90, how is this break released? This needs to be experimentally supported to justify the central conclusion of the paper.

I don't know if I would assume this break is actually released. It is certainly possible but it is also possible that non-yeast Ahsa1 has evolved to be recruited to Hsp90 in an entirely different way. The significance of the ATPase rate is not at all understood in terms of the client maturation cycle – which is of course the true function of Hsp90. There are many mutant forms of Hsp90 that have very high intrinsic (or Aha1-stimulated) ATPase rates that have dramatic client activation defects and poorly support viability in yeast. I think the central conclusions of the paper are three-fold and are supported by the data included. 1) That the 20 amino acid ICD limits ATPase stimulation, in part by interfering with the conserved NxNNWHW motif (which is essential for Aha1 function in yeast), 2) that the 20 amino acid ICD governs recruitment to Hsp90/client complexes, and 3) that the manner in which Ahsa1 is recruited to Hsp90 is different in non-yeast organisms. This is highly significant because almost everything that is known about Aha1 is based on studies with yeast cells and with yeast proteins.

2. the authors report point mutant effects in the 21-27 rescue segment.

a. These mutants are less relevant for supporting the central conclusion than mutations in residues 1-20 would be. Thus, which residues in residues 1-20 are crucial?

We did not carry out a comprehensive analysis of the 20 amino acids in this manuscript. I hope the reviewer would agree that this would constitute a completely different study. It is possible that the 20 amino acids could be post-translationally modified in a manner that alters its behaviour, that it interacts with another cellular factor we have not identified, or that it interacts directly with some part of Ahsa1 or Hsp90. We will be exploring these ideas in future work.

b. Why is Delta27 in some experiments more effective than wild type? This suggests that residues 21-27 are not important for stimulating the ATPase of Hsp90, which seem to contradict central assumptions of the authors.

I am assuming what is being referred to here is the appearance of the curves in Figure 4. We only added Ahsa1 up to 32 μM in our assays. Because the apparent affinity of the $\Delta 27$ construct was so much lower, the fit that resulted in the B_{max} calculation was not as accurate as those for the other constructs. I think, while not necessarily practical, if we had done one or two more concentrations (64 and 128 μM) of Ahsa1, the B_{max} would be very similar. However, if one considers the fact that many of our full length Ahsa1 constructs harbouring mutations in the NxNNWHW motif were worse than $\Delta 27$ illustrates one of our conclusions which is that the ICD inhibits ATPase stimulation by some other mechanism than simply neutralizing the NxNNWHW.

c. The mutations in residues 21-27 have rather mild effects, raising the question how important this region actually is. please clarify.

Many of the mutations have quite severe effects on ATPase stimulation but I would be reluctant to attribute this to 'importance' in a biological sense. As mentioned above, there are numerous Hsp90 mutants with very high ATPase rates that function very poorly biologically and many Hsp90 mutants with low rates that function very well biologically. The biochemical parameters we measured like apparent affinity for ATP and rate is better viewed as a readout of some specific step in the ATPase cycle. Future structural studies will be needed to determine this – preferably with non-yeast proteins.

3. what would be the proposed mechanism for auto inhibition?

a. the authors provide a structural; model, but this does not clarify this point. Which are the interactions the authors think are key for the proposed effects?

There are numerous mechanisms through which the ICD could be interfering with ATPase stimulation by Ahsa1. We have added a small section to the discussion to explore some of the possibilities but more structural work will be required to determine this.

b. Please annotate the relevant proteins Ana residues within the figure. At present the legend does not provide enough information to understand the figure.

We have tried to clarify the legend in this figure (9C) and have also labelled the W, H, and W residues in the picture. The orientation for the N residues in NxNN make it impossible to do the same individual labelling so we have added a colour coded label for the set.

Dear Paul,

Thank you for your patience while your revised manuscript was under review and thank you for providing feedback and a point-by-point response to the remaining concerns from Referee #3.

As discussed, please address all remaining concerns in a point-by-point response and please also address the following editorial points:

- 1) Please provide up to 5 keywords.
- 2) Please update the 'Conflict of interest' paragraph to our new 'Disclosure and competing interests statement'. For more information see <https://www.embopress.org/page/journal/14693178/authorguide#conflictsofinterest>
- 3) Please update the references to the alphabetical Harvard style. The abbreviation 'et al' should be used if more than 10 authors. You can download the respective EndNote file from our Guide to Authors https://endnote.com/style_download/embo-reports/
- 4) Please provide a complete author checklist, which you can download from our author guidelines (<<https://www.embopress.org/page/journal/14693178/authorguide>>). Please insert information in the checklist that is also reflected in the manuscript. The completed author checklist will also be part of the RPF.
- 5) Preprint citations: Please add the prefix 'preprint:' to the in-text citation and [PREPRINT] in the reference list. I.e., in-text citation (preprint: NAME1 et al, YEAR) and in reference list Author NAME1, Author NAME2, (YEAR) article title. bioRxiv doi: nnn [PREPRINT]
See also <https://www.embopress.org/page/journal/14693178/authorguide#referencesformat>
- 6) Please provide the figures at higher resolution.
- 7) Figure 3 has only one panel and I therefore suggest removing the label 'A'.
- 8) Generally, some of the figures contain only a small number of panels and you might want to consider combining some, e.g., Figure 7 and Figure 8.
- 9) Figure callouts are missing for the individual panels of Figure 5 and Figure 6.
- 10) Callouts for the EV figures need to be updated: we need "Figure EV1", "Figure EV2" etc. instead of "Figure S1", "Figure S2", etc.
- 11) Methods should be Materials and Methods
- 12) The manuscript sections should be in the following order: Title page - Abstract & Keywords - Introduction - Results - Discussion - Materials & Methods - Data Availability - Acknowledgments - Disclosure Statement & Competing Interests - References - Figure Legends - Expanded View Figure Legends.
- 13) Supporting figures need update in nomenclature (file names, legends, callouts): we need "Figure EV1", "Figure EV2" etc. instead of "Figure S1", "Figure S2", etc.
- 14) Source data: We perform a routine check on all quantitative source data. In this case we noticed that you have re-used control measurements to generate the graphs in Figure 2B and Figure 2C and in Figure S1 (see attached .xls sheet with color coded duplications). I assume that these measurements come from a multiplex experiment? Please clarify whether the controls were measured at the same time and on the same plate as all the test conditions and whether they are thus appropriate controls. Please also indicate this in the figure legend and the .xls file, i.e., that the data shown in Fig S1A and B comes from one experiment, using the same controls but that they are displayed in separate graphs etc.
- 15) Source data: We need separate .xls files per figure, uploaded as 1 file/folder per figure.
- 16) Western blot source data should be provided as image files within one folder instead of within an Excel file.
- 17) Source data for EV figures can be grouped into one folder and uploaded as zipped folder. Please do not forget to update the names from Figure Sx to Figure EVx.

18) Our production/data editors have asked you to clarify several points in the figure legends (see below). Please incorporate these changes in the manuscript and return the revised file with tracked changes with your final manuscript submission.

-Please note that a separate 'Data Information' section is required in the legends of figures 2b-e; 4a-f; 5a-d. [Data Information summarizes information that applies to several panels in the figure, e.g. "Data information: In (B-D), data are presented as mean {plus minus} SEM. *P{less than or equal to}0.05 (Student's t-test)."

- Please define the annotated p values ****/**/* in the legend of supplementary figures 2a-b; 4a-b; as appropriate.

- Please indicate the statistical test used for data analysis in the legends of supplementary figures 2a-b; 4a-b.

- Please note that in figures 2d-e; 5a-d; 6a-b; there is a mismatch between the annotated p values in the figure legend and the annotated p values in the figure file that should be corrected.

- Please note that information related to n is missing in the legends of figures 4a-f.

Although 'n' is provided, please describe the nature of entity for 'n' in the legends of figures 2b-e; 5a-d; 6a-b, supplementary figure 3a.

- Please note that the error bars are not defined in the legends of figures 2b-e; 3; 4a-f; 5a-d; 6a-b, supplementary figures 1a-d; 2a-b; 3a; 4a-b.

19) Finally, EMBO Reports papers are accompanied online by

A) a short (1-2 sentences) summary of the findings and their significance,

B) 2-3 bullet points highlighting key results and

C) a schematic summary figure that provides a sketch of the major findings (not a data image).

Please provide the summary figure as a separate file in PNG or JPG format at a size of 550x300-600 pixels (width x height).

Please note that the size is rather small and that text needs to be readable at the final size. Please send us this information along with the revised manuscript.

With kind regards,

Martina

Referee #2:

The authors have addressed my queries in a satisfactory manner.

The new data improve the study.

Referee #3:

The authors did not address some key concerns of this reviewer in the manuscript, or it had at least not been evident either they addressed these points. The authors should be encouraged to submit a revised version of their manuscript that addresses the concerns raised, together with a p-bp reply detailing how and where the changes had been made in the manuscript.

below I list the relevant issues from the point-by-point reply below, including my new comments between the lines.

Major concerns.

1. The authors claim that the N-terminal 20 residues act as autoinhibitor for Hsp90, suppress the ATPase and reduce affinity for Hsp90.

a. The authors need to determine the KiDs for AHSA1 binding to Hsp90, with and without the 20 N-terminal residues to substantiate this claim, via direct interaction measurements.

The apparent affinity we measured in our ATPase assays only describes the interaction that results in stimulation. However, since there are two binding sites for AhSA1 on every Hsp90 dimer, we can't comment on the affinity for the second site since it doesn't result in additional stimulation. We considered using SPR to address this concern but my work as a postdoc made me reluctant because the chip coupling can interfere with conformational changes in Hsp90 (or Aha1) that occur during the rather

complex binding process. Instead, we did ITC and, consistent with what we inferred from our ATPase data, we did not observe a large difference between Ahsa1 and Ahsa1 Δ 20 with respect to their affinity for Hsp90. The binding affinities also matched quite well with the apparent affinities we measured using the ATPase assay. We have included this new data as a supplemental figure (S3).

REVIEWER COMMENT

The new data provided in Fig S3 suggest a very mild change in affinity (KD change from 2 μ M to 3 μ M). This contradicts the claim that the 20 N-terminal residues would be an autoinhibitor.

- please adapt the claims accordingly and indicate in the p[BP] where and how this was done.

b. The authors need to clarify how the system should work - if the first 20 residues that act as autoinhibitor also reduce affinity for Hsp90, how is this break released? This needs to be experimentally supported to justify the central conclusion of the paper. I don't know if I would assume this break is actually released. It is certainly possible but it is also possible that non-yeast Ahsa1 has evolved to be recruited to Hsp90 in an entirely different way. The significance of the ATPase rate is not at all understood in terms of the client maturation cycle - which is of course the true function of Hsp90. There are many mutant forms of Hsp90 that have very high intrinsic (or Aha1-stimulated) ATPase rates that have dramatic client activation defects and poorly support viability in yeast. I think the central conclusions of the paper are three-fold and are supported by the data included. 1) That the 20 amino acid ICD limits ATPase stimulation, in part by interfering with the conserved NxNNWHW motif (which is essential for Aha1 function in yeast), 2) that the 20 amino acid ICD governs recruitment to Hsp90/client complexes, and 3) that the manner in which Ahsa1 is recruited to Hsp90 is different in non-yeast organisms. This is highly significant because almost everything that is known about Aha1 is based on studies with yeast cells and with yeast proteins.

COMMENT REVIEWER

How has this been adapted in the manuscript?

2. the authors report point mutant effects in the 21-27 rescue segment.

a. These mutants are less relevant for supporting the central conclusion than mutations in

residues 1-20 would be. Thus, which residues in residues 1-20 are crucial?

We did not carry out a comprehensive analysis of the 20 amino acids in this manuscript. I hope the reviewer would agree that this would constitute a completely different study. It is possible that the 20 amino acids could be post-translationally modified in a manner that alters its behaviour, that it interacts with another cellular factor we have not identified, or that it interacts directly with some part of Ahsa1 or Hsp90. We will be exploring these ideas in future work.

COMMENT REVIEWER

I do not agree with the authors that it would be a completely different study to identify the key element in residues 1-20. In fact, the authors make a bold claim that this region governs recruitment to Hsp90-client complexes. We now learn that change in affinity changes affinity very mildly (see Fig. S3 and above). Thus identification of key residues in the 1-20 region could help to support the claim, alternatively the authors should propose other approaches to support this claim.

c. The mutations in residues 21-27 have rather mild effects, raising the question how important this region actually is. please clarify.

Many of the mutations have quite severe effects on ATPase stimulation but I would be reluctant to attribute this to 'importance' in a biological sense. As mentioned above, there are numerous Hsp90 mutants with very high ATPase rates that function very poorly biologically and many Hsp90 mutants with low rates that function very well biologically. The biochemical parameters we measured like apparent affinity for ATP and rate is better viewed as a readout of some specific step in the ATPase cycle. Future structural studies will be needed to determine this - preferably with non-yeast proteins.

COMMENT REVIEWER:

How is: "The biochemical parameters we measured like apparent affinity for ATP and rate is better viewed as a readout of some specific step in the ATPase cycle." clarified in the revised manuscript?

3. what would be the proposed mechanism for auto inhibition?

a. the authors provide a structural; model, but this does not clarify this point. Which are the interactions the authors think are key for the proposed effects?

There are numerous mechanisms through which the ICD could be interfering with ATPase stimulation by Ahsa1. We have added a small section to the discussion to explore some of the possibilities but more structural work will be required to determine this.

COMMENT REVIEWER:

It its not fully clear which section in the discussion is referred to, but the point remains that this is insufficiently addressed.

Referee #3

The authors did not address some key concerns of this reviewer in the manuscript, or it had at least not been evident either they addressed these points. The authors should be encouraged to submit a revised version of their manuscript that addresses the concerns raised, together with a p-bp reply detailing how and where the changes had been made in the manuscript.

below I list the relevant issues from the point-by-point reply below, including my new comments between the lines.

Major concerns.

This is the original comment from the first round of review.

1. The authors claim that the N-terminal 20 residues act as autoinhibitor for Hsp90, suppress the ATPase and reduce affinity for Hsp90.
 - a. The authors need to determine the KiDs for Ahsa1 binding to Hsp90, with and without the 20 N-terminal residues to substantiate this claim, via direct interaction measurements.

This is our response from the first resubmission.

The apparent affinity we measured in our ATPase assays only describes the interaction that results in stimulation. However, since there are two binding sites for Ahsa1 on every Hsp90 dimer, we can't comment on the affinity for the second site since it doesn't result in additional stimulation. We considered using SPR to address this concern but my work as a postdoc made me reluctant because the chip coupling can interfere with conformational changes in Hsp90 (or Aha1) that occur during the rather complex binding process. Instead, we did ITC and, consistent with what we inferred from our ATPase data, we did not observe a large difference between Ahsa1 and Ahsa1 Δ 20 with respect to their affinity for Hsp90. The binding affinities also matched quite well with the apparent affinities we measured using the ATPase assay. We have included this new data as a supplemental figure (S3).

This is the concern raised in the second review.

REVIEWER COMMENT

The new data provided in Fig S3 suggest a very mild change in affinity (KD change from 2 μ M to 3 μ M). This contradicts the claim that the 20 N-terminal residues would be an autoinhibitor.
- please adapt the claims accordingly and indicate in the p[BP] where and how this was done.

We apologize if our original claim was not clear. The reviewer summarizes our claims as follows "The authors claim that the N-terminal 20 residues act as autoinhibitor for Hsp90, suppress the ATPase and reduce affinity for Hsp90." This is not an accurate summary of what we are claiming. Nowhere in the manuscript do we claim that the deletion of the 20 amino acid ICD reduces affinity for Hsp90. The autoinhibition we refer to is related to ATPase stimulation AFTER binding occurs. We show using two assays that there is no difference in affinity for Hsp90 upon deletion of the 20 amino acid ICD. We added the ITC experiments (direct measurements of affinity) to address this concern the first time but the same comment is being made in the second review. I think this might be a misunderstanding of what our original claims

were or of how we are using the term 'autoinhibition'. This seemed clear to the other two reviewers. It is difficult to point to where in our manuscript something wasn't said but I can point to the following statements where we say that deletion of the 20 amino acid ICD does not change binding affinity for Hsp90.

"Additionally, while the K_{app} of Ahsa1 for Hsp90 was unchanged after deletion of the ICD, further deletion of the NxNNWHW motif reduced the affinity of Ahsa1 for both Hsp90 α and Hsp90 β "

"This was particularly surprising for the Ahsa1 Δ 20 construct, as there was no difference in affinity for Hsp90 in our *in vitro* ATPase assays (**Figure 2D**)."

We do state that further deletion of the NxNNWHW motif reduces binding affinity for Hsp90 but this is a separate issue as this motif is not contained in the ICD.

This is the original comment from the first round of review.

b. The authors need to clarify how the system should work - if the first 20 residues that act as autoinhibitor also reduce affinity for Hsp90, how is this break released? This needs to be experimentally supported to justify the central conclusion of the paper.

This is our response from the first resubmission.

I don't know if I would assume this break is actually released. It is certainly possible but it is also possible that non-yeast Ahsa1 has evolved to be recruited to Hsp90 in an entirely different way. The significance of the ATPase rate is not at all understood in terms of the client maturation cycle - which is of course the true function of Hsp90. There are many mutant forms of Hsp90 that have very high intrinsic (or Aha1-stimulated) ATPase rates that have dramatic client activation defects and poorly support viability in yeast. I think the central conclusions of the paper are three-fold and are supported by the data included. 1) That the 20 amino acid ICD limits ATPase stimulation, in part by interfering with the conserved NxNNWHW motif (which is essential for Aha1 function in yeast), 2) that the 20 amino acid ICD governs recruitment to Hsp90/client complexes, and 3) that the manner in which Ahsa1 is recruited to Hsp90 is different in non-yeast organisms. This is highly significant because almost everything that is known about Aha1 is based on studies with yeast cells and with yeast proteins.

This is the concern raised in the second review.

COMMENT REVIEWER

How has this been adapted in the manuscript?

This concern appears to be more related to the misunderstanding of our claims of binding affinity for Hsp90. Again, we do not claim anywhere in the paper that the 20 amino acid ICD governs affinity for Hsp90. Our original response was an attempt to explain this but it appears to have been ineffective. It is still not clear to me what the reviewer think the 'central claim' of our paper is but I can try to explain it here. Our *in vitro* experiments show that deletion of the ICD does not alter binding affinity for Hsp90. However, deletion of this 20 amino acid section of Ahsa1 allows for more robust ATPase stimulation and/or nucleotide exchange (our previous work (Mercier 2019 Nat Comms) shows that Aha1 acts as both a stimulator and exchange factor for Hsp90 in yeast). An important aspect of the Hsp90 system in a biological/cellular

context is that co-chaperones (of which there are a dozen in yeast and several dozen in mammalian cells) are vastly substoichiometric to Hsp90. What this means is that mechanisms have evolved to regulate the recruitment of these co-chaperones to Hsp90 at key points in the cycle or in response to client requirements. If this weren't the case, it would be difficult to imagine how more than one type of co-chaperone could be recruited to a given Hsp90 dimer in the context of a single client activation cycle. Despite not altering the in vitro affinity of Aha1 for Hsp90, PTMs of Hsp90 such as phosphorylation of T22 and Y24, and SUMOylation of K178 (all yeast numbering) regulate the recruitment of Aha1 to Hsp90 in cells. This is why it is so significant that the 20 amino acid ICD appears to be regulating recruitment of Ahsa1 to Hsp90 in mammalian cells – which would constitute a fundamental difference with the yeast system in which Aha1 lacks an ICD. The only activity attributed to this portion of Ahsa1 in the mammalian system is the ability to bind to misfolded substrates. We added new data with the E67K mutant of Ahsa1 (which impairs binding to Hsp90 both in vitro and in vivo) to determine if the loss of interaction with Hsp90 would affect complex formation with the model client, glucocorticoid hormone receptor.

In terms of how this is supposed to work, we propose models in the discussion where the ICD could be responsible for the recruitment of Ahsa1 to Hsp90-client complexes in which the client was persistently misfolded and/or that the limits on ATPase stimulation mediated by the ICD could be alleviated through an interaction with a persistently misfolded client that is in complex with Hsp90. The relevant text from the discussion is below...

“Ahsa1 plays a role in the fate of important Hsp90 clients such as Tau and CFTR^{38, 39}. In the case of CFTR, cellular Ahsa1 levels appear to govern the balance between folding and degradation³⁸. This model is consistent with other chaperone cycles where attempts to fold will be made for a certain period of time or number of cycles (*i.e.* in the case of the calnexin/calreticulin cycle)⁵⁰. In the case of Hsp90 and Ahsa1, Ahsa1 could be recruited to switch the function of Hsp90 from folding to degradation. The previously identified ICD (that can interact with misfolded proteins) could provide the means for Ahsa1 to recognize Hsp90 complexes with persistently misfolded clients. Alternatively, Ahsa1 could be capable of recognizing clients for delivery to Hsp90. However, our co-IP data with Ahsa1^{E67K} (that cannot bind Hsp90) shows no interaction with GR suggesting that this is not the case for this client. That we did not recover Hsp90 or GR in complex with Ahsa1^{Δ20} suggests two possible scenarios for the effect of deletion of the ICD (**Figure 9B**) Loss of the ICD may prevent recruitment to the Hsp90/GR complex at all but another possibility is that the absence of the ICD results in such rapid ATP hydrolysis that complexes with Hsp90 and GR cannot be recovered. More work will be required to determine the mechanics and functional consequences of Ahsa1 interaction with Hsp90.”

This is the original comment from the first round of review.

2. the authors report point mutant effects in the 21-27 rescue segment.
 - a. These mutants are less relevant for supporting the central conclusion than mutations in residues 1-20 would be. Thus, which residues in residues 1-20 are crucial?

This is our response from the first resubmission.

We did not carry out a comprehensive analysis of the 20 amino acids in this manuscript. I hope the reviewer would agree that this would constitute a completely different study. It is possible that the 20 amino acids could be post-translationally modified in a manner that alters its behaviour, that it interacts with another cellular factor we have not identified, or that it interacts directly with some part of Ahsa1 or Hsp90. We will be exploring these ideas in future work.

This is the concern raised in the second review.

COMMENT REVIEWER

I do not agree with the authors that it would be a completely different study to identify the key element in residues 1-20. In fact, the authors make a bold claim that this region governs recruitment to Hsp90-client complexes. We now learn that change in affinity changes affinity very mildly (see Fig. S3 and above). Thus identification of key residues in the 1-20 region could help to support the claim, alternatively the authors should propose other approaches to support this claim.

The reviewer frames this again in the context of a claim we do not make. The data presented in Figure S3 does not result in new or surprising insight regarding the biophysical affinity of our Ahsa1 constructs for Hsp90. Rather, Figure S3 confirms what we state clearly in the manuscript and show clearly in Figure 2E, that the deletion of the 20 amino acids does not alter the affinity for Hsp90. The reviewer further suggests that our conclusion that the 20 amino acids is required for recruitment of Ahsa1 to Hsp90-GR complexes is bold and not supported by the data. I would again refer to the stoichiometry of co-chaperones to Hsp90. The extensive literature regarding the recruitment of co-chaperones to Hsp90 at specific points in the client activation cycle suggest that co-chaperones are only recruited to specific conformational intermediates of Hsp90 or under the control of PTMs in either Hsp90 or co-chaperones themselves. That the 20 amino acids ICD is playing such a role is supported by the data we present and in my opinion is not controversial for that reason. We added additional data with a mutant that blocks binding to Hsp90 (E67K) to show that this mutation results in the loss of complex formation with GR suggesting the model we propose in Figure 9B is reasonable. However, we also acknowledge other possibilities - specifically that the deletion of the ICD results in rapid resolution of the Hsp90-Ahsa1-GR complex because of the enhanced ATPase stimulation. We view this as a less likely explanation because the IPs are done in the presence of sodium molybdate which is well known to stabilize 'late' Hsp90 complexes. Nonetheless we acknowledge the possibility in the discussion.

I do not think that expressing and purifying 20 additional proteins for characterization is worthwhile and seems excessive to address this curiosity. Moreover, it would also involve testing 20 new mutants in the mammalian cell assays. Moreover, knowing the individual contributions of each residue would not suggest a mechanism for how this motif is exerting its effect in cells or in vitro. Without a structure of the human Ahsa1-Hsp90 complex, there is no way to attribute any individual contribution to a mechanism.

This is the original comment from the first round of review.

c. The mutations in residues 21-27 have rather mild effects, raising the question how important this region actually is. please clarify.

This is our response from the first resubmission.

Many of the mutations have quite severe effects on ATPase stimulation but I would be reluctant to attribute this to 'importance' in a biological sense. As mentioned above, there are numerous Hsp90 mutants with very high ATPase rates that function very poorly biologically and many Hsp90 mutants with low rates that function very well biologically. The biochemical parameters we measured like apparent affinity for ATP and rate is better viewed as a readout of some specific step in the ATPase cycle. Future structural studies will be needed to determine this - preferably with non-yeast proteins.

This is the concern raised in the second review.

COMMENT REVIEWER:

How is: "The biochemical parameters we measured like apparent affinity for ATP and rate is better viewed as a readout of some specific step in the ATPase cycle. " clarified in the revised manuscript?

The review initially comments on the mild effects of the point mutations we introduced in the NxNNWHW motif of Ahsa1. Most of these mutations actually have quite severe effects on ATPase stimulation (B_{max}) apparent binding affinity (K_{app}) and apparent K_m for ATP. My response in the first round of review was an attempt to explain why it is not responsible to attribute the changes in enzymatic and biophysical parameters to a specific aspect of the ATPase stimulation mechanism. We specifically add in the discussion the following...

"That mutations in the NxNNWHW impair ATPase stimulation by full length Ahsa1 to a greater degree than deletion of the entire ICD/NxNNWHW (in Ahsa1 $\Delta 27$) illustrates the fact that the ICD inhibits more than one facet of this process. If one assumes that the structure of the yeast Hsp90/Aha1 complex⁴⁹ is similar to that of the mammalian proteins, it is likely the ICD is located very near the Hsp90 N domains and could potentially interfere with N terminal dimerization, ATP lid dynamics, or strand exchange. Any of these inhibitory mechanisms would not be associated with Ahsa1 $\Delta 27$ which may explain its higher stimulation rate than some of the full-length NxNNWHW mutants."

...to acknowledge the many ways the NxNNWHW motif could stimulate ATP hydrolysis or nucleotide exchange. Outside of cryo-EM, we do not have the tools to interrogate this.

This is the original comment from the first round of review.

3. what would be the proposed mechanism for auto inhibition?

a. the authors provide a structural; model, but this does not clarify this point. Which are the interactions the authors think are key for the proposed effects?

This is our response from the first resubmission.

There are numerous mechanisms through which the ICD could be interfering with ATPase stimulation by Ahsa1. We have added a small section to the discussion to explore some of the possibilities but more structural work will be required to determine this.

This is the concern raised in the second review.

COMMENT REVIEWER:

It is not fully clear which section in the discussion is referred to, but the point remains that this is insufficiently addressed.

We discuss this in the first paragraph of the discussion at the end of which we added the text in italics...

The 20 amino acid sequence at the N-terminus of Ahsa1, despite its conservation in virtually every member of this cochaperone family, has been largely overlooked because it is absent in yeast Aha1. We report here that this sequence autoinhibits ATPase stimulation, in part by partially masking the NxNNWHW motif (**Figure 9A**). More specifically, the ICD interferes with the WHW portion of the NxNNWHW motif but must also interfere with either another part of Ahsa1 or with Hsp90 directly. Deletion of these 20 amino acids results in a significant increase in Hsp90 ATPase stimulation. The only previous reports on this sequence suggested that it can interact with misfolded proteins directly and prevent aggregation^{43, 44}. In one of these studies, a 22 amino acid deletion did not significantly increase ATPase stimulation of Hsp90. However, this could have been because part of the NxNNWHW was deleted in their construct. Indeed, we showed that the N21A substitution in Ahsa1 impaired Hsp90 ATPase stimulation. It is possible that the improvement in ATPase stimulation resulting from the deletion of the first twenty amino acids could have been masked by the deleterious effect of deleting the first two amino acids of the NxNNWHW motif. Additionally, we measured a far greater Ahsa1-stimulated Hsp90 ATPase rate (*i.e.* 3-5 fold greater) in our experiments. Regardless of the mechanism for this discrepancy, the previously reported autonomous chaperone function of this region of Ahsa1 may shed light on how Ahsa1 regulates Hsp90 function. *That mutations in the NxNNWHW impair ATPase stimulation by full length Ahsa1 to a greater degree than deletion of the entire ICD/NxNNWHW (in Ahsa1^{Δ27}) illustrates the fact that the ICD inhibits more than one facet of this process. If one assumes that the structure of the yeast Hsp90/Aha1 complex⁴⁹ is similar to that of the mammalian proteins, it is likely the ICD is located very near the Hsp90 N domains and could potentially interfere with N terminal dimerization, ATP lid dynamics, or strand exchange. Any of these inhibitory mechanisms would not be associated with Ahsa1^{Δ27} which may explain its higher stimulation rate than some of the full-length NxNNWHW mutants.*

Manuscript number: EMBOR-2023-57657V3

Title: Recruitment of Ahsa1 to Hsp90 is regulated by a conserved peptide that inhibits ATPase stimulation

Author(s): Solomon Hussein, Rakesh Bhat, Michael Overduin, and Paul LaPointe

Dear Paul,

Thank you for your patience while we have editorially reviewed your revised manuscript. All files are in good order now, but I have a few minor requests and since these require the upload of new figure files, I will 'open' the manuscript for you so that you can perform the uploads.

I am writing with an 'accept in principle' decision, which means that I will be happy to accept your manuscript for publication once these minor issues/corrections have been addressed, as follows.

- I still consider the figure resolution suboptimal. E.g., Figure 1: if I print the figure at final size (180 mm for a two-column wide figure) the text cannot be read. Even if I assume that not many readers print articles these days and view it online instead, the Figure looks OK at 100% zoom but becomes blurry when I zoom in. Our Figure Guidelines recommend a resolution of 300 pixels per inch at the actual/final print size. Please carefully review the figures and check what it looks like when you print it at final size (87 mm for one-column, 180 mm for two-columns) and when you try to zoom in.

- I corrected three minor things in the manuscript text (attached).

- The synopsis image also looks blurry at final size of 550 pixels width. Please check and correct text and graphics.

- Please also review the minor changes to the synopsis text (attached).

Once you have made these minor revisions, please use the following link to submit your corrected manuscript:

Link Not Available

If all remaining corrections have been attended to, you will then receive an official decision letter from the journal accepting your manuscript for publication in the next available issue of EMBO reports. This letter will also include details of the further steps you need to take for the prompt inclusion of your manuscript in our next available issue.

Thank you for your contribution to EMBO reports.

Kind regards,

Martina

All editorial and formatting issues were resolved by the authors.

Dr. Paul LaPointe
University of Alberta
Cell Biology
5-51 Medical Sciences Building
Edmonton, Alberta T6G2H7
Canada

Dear Paul,

Thank you for implementing the last minor changes. Figure 1 has indeed a better resolution now. I am thus very pleased to accept your manuscript for publication in the next available issue of EMBO reports. Thank you for your contribution to our journal.

Best regards,

Martina
